# Synthesizable Molecular Generation
# via Soft-constrained GFlowNets with Rich Chemical Priors

**Hyeonah Kim** [1 2]   **Minsu Kim** [1 3]   **Celine Roget** [1 2]   **Dionessa Biton** [1 2]   **Louis Vaillancourt** [2]   **Yves V. Brun** [2 4]
**Yoshua Bengio** [1 2]   **Alex Hernandez-Garcia** [1 2 5]

## Abstract

The application of generative models for experimental drug discovery campaigns is severely limited by the difficulty of designing molecules *de novo* that can be synthesized in practice. Previous works have leveraged Generative Flow Networks (GFlowNets) to impose hard synthesizability constraints through the design of state and action spaces based on predefined reaction templates and building blocks. Despite the promising prospects of this approach, it currently lacks flexibility and scalability. As an alternative, we propose S3-GFN, which generates synthesizable SMILES molecules via simple soft regularization of a sequence-based GFlowNet. Our approach leverages rich molecular priors learned from large-scale SMILES corpora to steer molecular generation towards high-reward, synthesizable chemical spaces. The model induces constraints through off-policy replay training with a contrastive learning signal based on separate buffers of synthesizable and unsynthesizable samples. Our experiments show that S3-GFN learns to generate synthesizable molecules ($\geq 95\%$) with higher rewards in diverse tasks.

## 1. Introduction

The design of small molecules with targeted properties is a central problem in chemistry and plays a key role in drug discovery. Recent advances in generative modeling have enabled *de novo* molecular design as a data-driven alternative. In particular, probabilistic frameworks such as Generative Flow Networks (GFlowNets; Bengio et al., 2023) are designed to sample diverse candidates in proportion to an unnormalized reward, naturally supporting multi-modal objectives common in drug discovery (Bengio et al., 2021; Jain et al., 2022; 2023; Kim et al., 2024a). However, evaluations in this area often focus on computational benchmarks alone, overlooking practical considerations required for experimental validation. As a result, many molecules predicted to have high scores of a target property are chemically unstable or unsynthesizable, limiting their utility in wet-lab settings (Gao & Coley, 2020; Papidocha et al., 2026).

To address synthesizability, recent work has proposed reaction-based generation frameworks. These methods formulate generation as a Markov Decision Process (MDP) where molecules are constructed by selecting predefined reaction templates and building blocks sequentially (Koziarski et al., 2024; Cretu et al., 2025; Seo et al., 2025). While this guarantees the existence of a synthesis pathway, such formulations introduce combinatorial action spaces that scale poorly with library size (Guo et al., 2025). In addition, these MDPs encode a fixed notion of synthesizability, making it difficult to accommodate alternative structural constraints or evolving experimental requirements. Also importantly, reliance on specialized reaction-based MDPs limits the ability to leverage rich chemical priors learned by foundation models pretrained on large chemical language corpora. Large-scale SMILES datasets—for instance, PubChem (Kim et al., 2025c), ZINC (Tingle et al., 2023), and ChEMBL (Mendez et al., 2019)—are far more abundant and transferable, whereas synthesis-pathway data is expensive to obtain and tightly coupled to specific reaction templates and building-block libraries.

In this paper, we propose **Synthesizable SMILES via Soft-constrained GFlowNet (S3-GFN)**, a molecular generation framework that induces synthesizability through distributional post-training rather than hard-coded reaction rules. S3-GFN combines off-policy GFlowNet training with soft constraint regularization, leveraging rich chemical priors learned from large-scale SMILES data. Our key idea is to model synthesizability as a soft constraint learned via contrastive learning and integrate it into the GFlowNet post-training objective. This regularization operates at the distributional level: probability mass is reweighted within the synthesizable region, while unsynthesizable regions are ex-

---

[1]Mila - Quebec AI Institute [2]Université de Montréal [3]KAIST [4]Institut Courtois d'innovation biomédicale [5]Institut Courtois. Correspondence to: Hyeonah Kim <hyeonah.kim@mila.quebec>, Alex Hernandez-Garcia <alex.hernandez-garcia@mila.quebec>.

*Proceedings of the 43ʳᵈ International Conference on Machine Learning*, Seoul, South Korea. PMLR 306, 2026. Copyright 2026 by the author(s).

plicitly suppressed through a contrastive auxiliary loss. By decoupling constraint enforcement from scalar rewards, this design avoids the optimization conflicts commonly observed in standard soft-regularization approaches like reward shaping.

Specifically, we initialize S3-GFN using GP-MolFormer (Ross et al., 2025), a pretrained SMILES language model, and optimize an amortized posterior via GFlowNet post-training. To implement the distributional regularization, we leverage the off-policy nature of GFlowNets by maintaining two separate replay buffers of synthesizable (i.e., positive) and unsynthesizable (i.e., negative) samples. During replay training, we introduce a contrastive auxiliary loss that penalizes the relative likelihood of negative trajectories against positive ones, explicitly suppressing probability mass outside the synthesizable region. In turn, the GFlowNet objective continues to learn the reward distribution within the synthesizable space. Figure 1 illustrates the overview of the method. The off-policy framework further enables external search operators to refine the replay buffers, such as generating negative samples via local mutations to strengthen the regularization, or generating high-reward positive samples via genetic search (Kim et al., 2024a).

Empirically, S3-GFN achieves high synthesizability rates, over 95%, while maintaining competitive optimization performance across diverse molecular design tasks, outperforming reaction-based GFlowNets despite operating on a simpler sequence-based MDP. We further observe that the proposed method enables rapid realignment of the sampling distribution under evolving constraints, requiring only a small number of additional training steps and exhibiting more stable behavior than reward shaping. Moreover, the ability to incorporate off-policy samples from external search operators enables S3-GFN to attain the strongest performance in sample-limited benchmarks. Together, these results show that combining rich chemical priors with constraint-aware distributional post-training enables effective, flexible, and scalable generation of synthesizable molecules across diverse design objectives.

## 2. Background and Related Work

### 2.1. Generative Flow Networks

Generative Flow Networks (GFlowNets or GFN) are an off-policy reinforcement learning (RL) method for amortized inference, which can learn probabilistic models $p_\theta(x)$ that sample proportionally to an unnormalized density or reward $R(x)$, that is $p_\theta(x) \propto R(x)$ (Bengio et al., 2021; 2023). GFlowNets model a sequential decision process represented as a trajectory $\tau = (s_0 \to \cdots \to s_n = x)$ from an initial state $s_0$ to a terminal state $s_n$, which represents the output solution $x$. As in RL, the set of states, of valid actions

between states and the reward, define a Markov decision process (MDP).

GFlowNets define two major distributions: a forward transition probability $P_F(\tau)$, modeling a distribution over trajectories, and a backward transition probability $P_B(\tau \mid x)$, modeling a distribution over backward trajectories given $x$. Both distributions are represented as compositions of state transition policies for the forward and backward processes:

$$P_F(\tau) = \prod_{t=0}^{n-1} P_F(s_{t+1} \mid s_t),$$

$$P_B(\tau \mid x) = \prod_{t=1}^{n} P_B(s_{t-1} \mid s_t).$$

**Trajectory balance** GFlowNets are trained to learn forward (and optionally backward) state transition policies using deep network parameterization $\theta$ for amortized inference. This learning can be formulated as a constraint-matching problem. Trajectory balance (TB; Malkin et al., 2022) is a widely used objective that enforces a global constraint over full trajectories. Other variants include sub-trajectory balance (SubTB; Madan et al., 2023), which enforces sub-global constraints, as well as detailed balance (DB; Bengio et al., 2023) and flow matching (Bengio et al., 2021), which impose local constraints. The TB objective is expressed as

$$\mathcal{L}_{\text{TB}}(\tau) = \left( \log \frac{Z_\theta P_F(\tau; \theta)}{R(x) P_B(\tau \mid x; \theta)} \right)^2. \quad (1)$$

By minimizing the TB loss to zero over all trajectories $\tau \in \mathcal{T}$, GFlowNet training guarantees correct inference, $p(x) = \sum_{\tau \to x} P_F(\tau) \propto R(x)$. The backward policy $P_B$ can be either trainable or fixed to a predefined distribution (e.g., uniform). In sequence generation (e.g., SMILES generation), the backward distribution is deterministic (i.e., $P_B(\tau \mid x) = 1$ if $\tau \to x$), so the TB objective reduces to Path Consistency Learning (PCL; Nachum et al., 2017), as shown by Deleu et al. (2024) and Tiapkin et al. (2024).

**Relative trajectory balance** GFlowNet training can also be used as a post-training method when a pretrained prior model $p^{\text{prior}}(x) = \sum_{\tau \to x} P_F^{\text{prior}}(\tau)$ is available, as proposed by Venkatraman et al. (2024) and later applied to molecular generation by Pandey et al. (2025). In this setting, the prior may correspond to a large pretrained model, such as a SMILES-based language model. Post-training GFlowNets aims to perform amortized posterior inference, $p^{\text{post}}(x) \propto R(x)p^{\text{prior}}(x)$. To achieve this, we can use a variation of the TB objective called relative trajectory balance (RTB) as

$$\mathcal{L}_{\text{RTB}}(\tau) = \left( \log \frac{Z_\theta P_F^{\text{post}}(\tau; \theta)}{R(x) P_F^{\text{prior}}(\tau)} \right)^2. \quad (2)$$

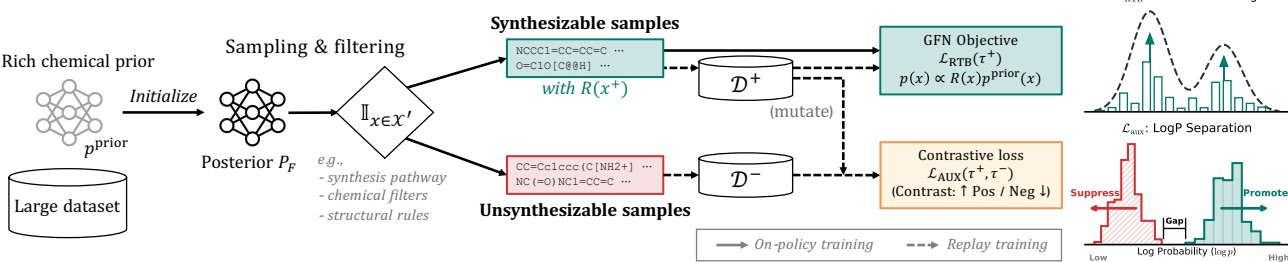

*Figure 1.* **Overview of Synthesizable SMILES via Soft-constrained GFN (S3-GFN).** A pretrained SMILES prior provides chemical plausibility and is continuously referenced through RTB. On-policy updates apply RTB using positive samples only, while replay updates introduce a contrastive auxiliary loss that separates positive and negative samples, while preserving shared substructures.

## 2.2. Reaction-based vs. Sequence-based Generation

The flexibility of the GFlowNets framework allows for different choices of MDPs to define the process by which molecules are generated. In this section, we discuss two MDP choices for molecular generation: reaction-based and sequence-based MDPs. We further discuss fragment-based MDP in Appendix B.2.

**Reaction-based MDPs** In reaction-based formulations, molecules are generated through synthetic pathways. A state $s_t$ represents an intermediate molecule obtained by applying a sequence of chemical reactions along a synthesis pathway. The initial state $s_0$ is empty, and a terminal state yields a final molecule $x$ as the product of a completed synthetic route. Let $\mathcal{M}$ denote a set of available molecular building blocks (chemical compounds used as reactants), and let $\mathcal{R} = \mathcal{R}_{\text{uni}} \cup \mathcal{R}_{\text{bi}}$ denote sets of unimolecular and bimolecular reaction templates. The possible actions are

$$
a_t = \begin{cases}
b \in \mathcal{M}, & \text{(init)} \\
r \in \mathcal{R}_{\text{uni}}, & \text{(unimolecular)} \\
(r, b), r \in \mathcal{R}_{\text{bi}}, b \in \mathcal{M}_r, & \text{(bimolecular)} \\
\texttt{stop}, &
\end{cases}
$$

where $\mathcal{M}_r$ is the subset of building blocks compatible with reaction template $r$. Therefore, the action space is state-dependent and restricted to chemically valid operations. Notably, action space scales combinatorially, resulting in a vast number of potential actions (e.g., considering 105 reactions with 200K building blocks).

**Sequence-based MDPs** In sequence-based formulations, molecule generation proceeds by the sequential construction of a molecular string representation, such as the Simplified Molecular-Input Line-Entry System (SMILES) strings (Weininger, 1988). A state $s_t$ represents a partial token sequence (i.e., a prefix of a SMILES string) generated up to step $t$, where the initial state corresponds to an empty string. Let $\mathcal{V}$ denote a fixed vocabulary of SMILES tokens. At each

state $s_t$, an action $a_t \in \mathcal{V}$ appends a token to the current prefix, deterministically producing the next state $s_{t+1}$.

Reaction-based and sequence-based MDPs present complementary trade-offs for molecular generation. Reaction-based MDPs offer chemically grounded and causally interpretable state–action representations, enforcing synthesizability through hard constraints since each molecule corresponds to a valid synthetic route. However, they rely on large state-dependent action spaces and are limited by predefined reaction templates and building block libraries. In contrast, sequence-based MDPs operate over a fixed token vocabulary, making them simpler to implement and well suited to language models pretrained on large-scale molecular datasets, but without guarantees of synthesizability. Addressing this limitation—inducing synthesizability in sequence-based molecular generation while preserving modeling flexibility—is the central focus of this work.

## 2.3. Synthesizable Molecule Generation

This section reviews representative approaches for incorporating synthesizability into molecular generation. Reaction-based generation has been explored across multiple generative frameworks, including genetic algorithms (e.g., SYN-OPSIS (Vinkers et al., 2003), SynNet (Gao et al., 2022b), SynGA (Lo et al., 2025)), tree search (Swanson et al., 2024), RL (Gottipati et al., 2020; Horwood & Noutahi, 2020; Swanson et al., 2025), and projection (Luo et al., 2024; Gao et al., 2024; Lee et al., 2026). More recently, GFlowNets have been combined with reaction-based generation to improve diversity and exploration under synthesis constraints, including RGFN (Koziarski et al., 2024), SynFlowNet (Cretu et al., 2025), and RxnFlow (Seo et al., 2025). Gaiński et al. (2025) and Shen et al. (2025) extend these works to synthesis cost-aware generation and 3D molecular design.

An alternative line of work promotes synthesizability through the training objective rather than the generation process. Guo & Schwaller (2025) and Guo et al. (2025) guide SMILES-based generation using retrosynthesis-driven reward shaping with a policy-gradient RL, retaining architec-

tural flexibility but entangling constraint enforcement with scalar reward optimization. For a comprehensive review, we refer readers to Papidocha et al. (2026).

## 3. Method

In this section, we present S3-GFN, a training framework for synthesizable molecular generation that jointly optimizes task-specific rewards (e.g., binding affinity) and synthesizability. The framework builds on a pretrained SMILES language model as a chemical prior and applies GFlowNet-based post-training with a soft, distributional regularization objective, as illustrated in Figure 1. Leveraging the off-policy nature of GFlowNet training, S3-GFN enables controlled use of both synthesizable and unsynthesizable samples via replay-buffer management, facilitating stable optimization of reward and constraint objectives.

---

**Algorithm 1** Synthesizability regularized post-training

---

**Require:** A prior $p^{\text{prior}}(\cdot)$, reward $R(x)$, synthesizable molecular space $\mathcal{X}'$, auxiliary coefficient $\alpha$, maximum iteration $N_{\text{iter}}$, and batch size $B$
1: Initialize $P_F \leftarrow p^{\text{prior}}$ {*warm-start from prior*}
2: Initialize $\log Z_\theta \leftarrow 0$
3: Initialize buffers $\mathcal{D}^+$ and $\mathcal{D}^-$
4: **for** $t = 1$ **to** $N_{\text{iter}}$ **do**
    *(A) On-policy sampling*
5:    Sample $\{\tau_i\}_{i=1}^B \sim P_F^{\text{post}}(\cdot; \theta)$
6:    $\mathcal{B}_t^+ \leftarrow \emptyset$
7:    **for** $i = 1$ **to** $B$ **do**
8:        $x_i \leftarrow s_n^{(i)}$ {$\tau_i = (s_0^{(i)} \rightarrow s_1^{(i)} \rightarrow \cdots \rightarrow s_n^{(i)})$}
9:        **if** $\mathbb{I}[x_i \in \mathcal{X}'] = 1$ **then**
10:        $\mathcal{B}_t^+ \leftarrow \mathcal{B}_t^+ \cup \{(\tau_i, R(x_i))\}$
11:        **else**
12:        $\mathcal{D}^- \leftarrow \mathcal{D}^- \cup \{\tau_i\}$
13:        **end if**
14:    **end for**
15:    $\mathcal{D}^+ \leftarrow \mathcal{D}^+ \cup \mathcal{B}_t^+$
    *(B) On-policy RTB update (positive samples only)*
16:    Update $\theta$ by minimizing $\mathcal{L}_{\text{RTB}}(\tau)$ over $\tau \in \mathcal{B}_t^+$
    *(C) Replay-based constraint separation*
17:    Sample $\tau^+ \sim \mathcal{D}^+$ and $\tau_{\text{buf}}^- \sim \mathcal{D}^-$
18:    (Optional) $\tau_{\text{mut}}^- \leftarrow \texttt{Mutate}(x^+)$
19:    Update $\theta$ by minimizing

$$\mathcal{L}_{\text{RTB}}(\tau^+) + \alpha \mathcal{L}_{\text{aux}}(\tau^+, \tau_{\text{buf/mut}}^-)$$

20: **end for**

---

**Problem definition** We pose the problem of molecular design as *de novo* SMILES generation under synthesizability constraints, without introducing specific MDPs tailored into constraints. Importantly, we assume access to a rich pretrained SMILES prior, trained on large commercial or curated datasets consisting almost entirely of synthesizable

molecules.[1] While such priors are not optimized for our task-specific objective, they provide a strong inductive bias toward chemically valid and synthesizable structures. Furthermore, we assume we have access to a method to determine whether a molecule $x \in \mathcal{X}$ satisfies the synthesizability constraint, that is $x \in \mathcal{X}' \subset \mathcal{X}$. For example, we define $\mathcal{X}'$ as the set of molecules possessing at least one valid synthetic route by using a heuristic retrosynthesis search procedure following Seo et al. (2025). Given a pretrained generator with a prior distribution $p^{\text{prior}}(x)$ and a task-specific score $R(x)$, our goal is to train a posterior

$$p(x) \propto R(x) p^{\text{prior}}(x) \mathbb{I}[x \in \mathcal{X}'].$$

**Overview** We approximate this constrained distribution through distributional post-training. As illustrated in Algorithm 1, our training procedure consists of two phases. During on-policy training (Section 3.1), we apply RTB in Equation (2) to positive samples ($x^+ \in \mathcal{X}'$) only, leveraging the pretrained prior to perform posterior reweighting within the synthesizable space $\mathcal{X}'$. During replay training (Section 3.2), we introduce a contrastive auxiliary loss that separates positive and negative (i.e., $x^- \in \mathcal{X} \setminus \mathcal{X}'$) molecules. Along with RTB on positive replayed samples, the auxiliary loss explicitly suppresses negative regions outside $\mathcal{X}'$.

### 3.1. On-policy Training with Positive Samples

At each iteration, we sample trajectories $\tau \sim P_F(\cdot)$, and the generated molecules are labeled as positive if $x \in \mathcal{X}'$ and negative otherwise. Here, $x$ is the final state $s_n$ in $\tau$.

**Positive-only on-policy training with RTB** For feasible trajectories, we compute rewards with $R(x)$ and denote $\mathcal{B}_t^+ = \{(\tau_i, R(x_i)) : x_i \in \mathcal{X}'\}$ the set of positive (synthesizable) trajectories sampled on-policy at iteration $t$. During on-policy training, we optimize the RTB objective defined in Equation (2) exclusively to feasible trajectories, that is

$$\mathcal{L}_{\text{RTB}}^+ = \frac{1}{|\mathcal{B}_t^+|} \sum_{\tau \in \mathcal{B}_t^+} \mathcal{L}_{\text{RTB}}(\tau). \tag{3}$$

Applying RTB to feasible trajectories focuses the posterior reweighting within the feasible region $\mathcal{X}'$ only. However, since this update does not explicitly control the probability mass outside $\mathcal{X}'$ the model may still assign non-negligible likelihood to negative molecules (see Figure 2).

### 3.2. Replay Training with Contrastive Auxiliary Loss

To further discourage the generation of unsynthesizable molecules, we introduce a contrastive auxiliary regulariza-

---

[1]This has become common practice in molecular generation, with open-source pretrained SMILES generative models such as GP-MoLFormer (Ross et al., 2025) and MegaMolBART (NVIDIA, 2021) publicly available.

tion loss that is applied only during replay training. By constructing replay mini-batches with balanced numbers of synthesizable (i.e., positive) and unsynthesizable (i.e., negative) trajectories, this auxiliary objective enables stable suppression of unsynthesizable regions—an outcome that is difficult to achieve with purely on-policy sampling. Importantly, the auxiliary loss explicitly induces constraint satisfaction without entangling it with reward optimization, in contrast to naive reward-shaping (RS) approaches.

**Off-policy contrastive replay buffer**  We maintain two replay buffers: a synthesizable buffer $\mathcal{D}^+$ and an unsynthesizable buffer $\mathcal{D}^-$. During replay updates, a mini-batch of positive samples $\mathcal{B}^+$ is drawn from $\mathcal{D}^+$ using reward-prioritized sampling, while a mini-batch of negative samples $\mathcal{B}^-$ is drawn from $\mathcal{D}^-$ uniformly. In practice, we set $|\mathcal{B}^+| = |\mathcal{B}^-| = B$, where $B$ is on-policy sample batch size.

Beyond storing trajectories encountered during training, the off-policy replay framework allows controlled manipulation of buffer contents. In particular, synthesizable molecules in $\mathcal{D}^+$ can be locally mutated to generate additional negative samples, which are added to $\mathcal{D}^-$. For a given positive sample $x^+$, we generate $x^-_{\mathrm{mut}} = \mathtt{Mutate}(x^+)$, using graph-based genetic operations from Jensen (2019), and obtain the corresponding trajectory $\tau^-_{\mathrm{mut}}$ directly in the SMILES space. On the other hand, auxiliary search procedures such as genetic search (Kim et al., 2024a) can be applied within $\mathcal{D}^+$ to discover high-reward synthesizable molecules, which are then reused during replay training via reward-prioritized sampling.

**Auxiliary loss**  We define the auxiliary loss as

$$\mathcal{L}_{\mathrm{aux}} = - \sum_{\tau \in \mathcal{B}^+} \log \frac{\exp(s(\tau))}{\exp(s(\tau)) + \sum_{\tau' \in \mathcal{B}^-} \exp(s(\tau'))},$$
(4)

where $s(\tau) = \log P_F(\tau)$ denotes the sequence-level score assigned by the forward policy, thus $\exp(s(\tau)) = P_F$. In practice, when mutation-based negatives are used, the auxiliary loss is computed separately and summed with the buffer-based term. With the auxiliary loss, we define the replay training loss for as

$$\mathcal{L}_{\mathrm{replay}} = \frac{1}{|\mathcal{B}^+|} \sum_{\tau \in \mathcal{B}^+} \mathcal{L}_{\mathrm{RTB}}(\tau) + \alpha \mathcal{L}_{\mathrm{aux}}.$$
(5)

We apply the auxiliary loss only during replay training. As training progresses and synthesizability improves, on-policy minibatches contain a highly variable (often vanishing) number of negative samples, making the auxiliary loss ill-defined and unstable. Replay training provides a consistent batch composition with controlled access to negative samples, enabling stable optimization of the auxiliary regularization.

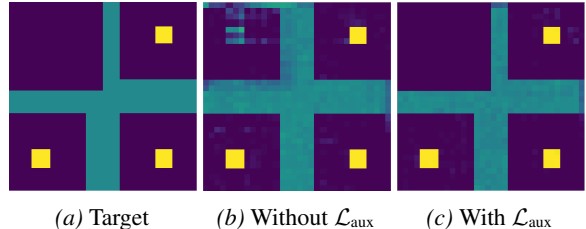

*(a)* Target          *(b)* Without $\mathcal{L}_{\mathrm{aux}}$          *(c)* With $\mathcal{L}_{\mathrm{aux}}$

*Figure 2.* **Deceptive 2D grid world with feasibility constraints.** (a) Target distribution, where black cells denote infeasible states and colors indicate reward levels. (b) Learned sampling distribution using feasible-only training, and (c) learned sampling distribution with the auxiliary contrastive loss $\mathcal{L}_{\mathrm{aux}}$.

## 4. Proof of Concept: 2D Grid World

In this section, we use a synthetic scenario to provide a proof-of-concept analysis of how constraint-aware post-training shapes the learned sampling distribution and to illustrate the role of the auxiliary contrastive loss in suppressing infeasible regions.

**Setup**  As illustrated in Figure 2a, we study a two-dimensional grid sampling space $\mathcal{X}$ with high-reward modes in the corners, shown in yellow, surrounded by low-reward regions, shown in purple, that hinder exploration. This setting has been previously used by Kim et al. (2025b). To simulate structural constraints, we set the upper-left quadrant as infeasible—trajectories terminating here are considered negative. We use TB in Equation (1) with positive samples only, and ablate the contrastive auxiliary loss term in replay training. The coefficient $\alpha$ is set to 0.01.

**Results**  Figure 2b shows that training with feasible samples alone assigns non-negligible probability mass to infeasible regions. Without explicit learning signal about the negative region, the model extrapolates the existing symmetry and assigns non-zero probability mass to unseen negative samples. In contrast, Figure 2c shows that incorporating the contrastive loss effectively suppresses the negative region, by explicitly penalizing the negative trajectories relative to positive counterparts. We conduct further study on the coefficients $\alpha$; see Figure 7 in Appendix C.1.

## 5. Synthesizable Molecular Generation

The experiments evaluate whether S3-GFN—soft-constraint handling via GFlowNet post-training, combined with rich pretrained SMILES priors—is effective at generating diverse, synthesizable molecules with high rewards. The code is available at https://github.com/hyeonahkimm/s3gfn.

**Common setup**  Across all experiments, we employ GP-MolFormer (Ross et al., 2025),[2] a chemical language model

---

[2]https://huggingface.co/ibm-research/GP-MoLFormer-Uniq

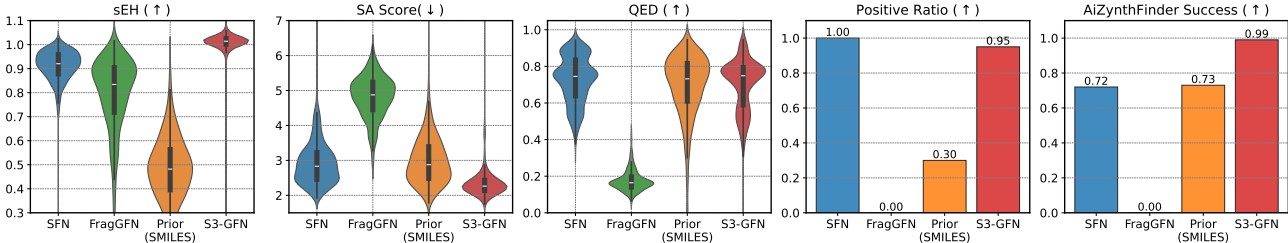

*Figure 3.* **Comparison over different MDPs on sEH.** While SFN guarantees 100% validity on the generation constraints (Positive Ratio), S3-GFN achieves higher success on AiZynthFinder and discovers candidates with consistently higher sEH scores.

(a) `O=C(CNC(=O)c1ccc2c(c1)CCC2)Nc1ccc2c3c(cccc13)CC2`

reward: 1.074

(b) `O=C(NCc1ccc2c(c1)CNC2)c1cc(-c2ccccc2)co1`

reward: 1.063

*Figure 4.* **Example of synthetic pathway of Top-2 candidates under given reaction $\mathcal{R}$ and building block $\mathcal{M}$.** With a high positive ratio, our generated molecules have valid synthetic pathways.

trained on large-scale molecular datasets as a pre-trained model to be fine-tuned for specific tasks. Following prior GFlowNet works on drug discovery, we define the target sampling distribution as $p(x) \propto p^{\mathrm{prior}}(x) \exp(-\mathcal{E}(x))$, where $\mathcal{E}(x) = -r(x)$ and $r(x)$ is the task-specific property score to be maximized. Notably, we consider molecule positive ($x \in \mathcal{X}'$) if a valid synthetic pathway exists given a predefined set of reaction templates $\mathcal{R}$ and building blocks $\mathcal{M}$. By default, we use the 105 reaction templates used in SynFlowNet and the Enamine Stock library for the building block set, with a maximum of three synthesis steps. In Equation (5), we fix the auxiliary loss coefficient to $\alpha = 10^{-3}$. In addition, we independently assess the synthesizability of generated molecules using AiZynthFinder (Genheden et al., 2020). This tool performs retrosynthesis planning based on the USPTO reaction (Lowe, 2017), serving as a metric for synthesizability (Cretu et al., 2025; Seo et al., 2025).

## 5.1. Main Results

We first compare SMILES-based generation with hard-coded reaction-based generation process on a proxy-based optimization and structure-based drug design tasks. Across all evaluated tasks, our approach achieves higher average rewards while maintaining strong synthesizability, despite operating on the simpler sequence-based MDP. This indicates that high-quality, synthesizable molecules can be found with a sequence-based GFlowNet, bypassing the need for more complicated reaction-based MDPs.

*Table 1.* **Results on sEH.** Mean and standard deviations are reported with three independent runs; see Table 6 for the full results.

| Method | Positive (↑) | Pos. Top100 sEH (↑) | sEH (↑) | Diversity (↑) |
|---|---|---|---|---|
| FragGFN | 0.0 | - | $0.790 \pm 0.002$ | $0.822 \pm 0.010$ |
| SFN | **1.0** | $0.989 \pm 0.005$ | $0.907 \pm 0.006$ | **0.801** $\pm 0.022$ |
| Prior | $0.299 \pm 0.014$ | $0.628 \pm 0.008$ | $0.482 \pm 0.003$ | $0.877 \pm 0.001$ |
| RTB + RS | $0.987 \pm 0.003$ | $1.040 \pm 0.001$ | $1.003 \pm 0.001$ | $0.765 \pm 0.002$ |
| S3-GFN | $0.945 \pm 0.009$ | **1.043** $\pm 0.001$ | **1.009** $\pm 0.000$ | $0.764 \pm 0.000$ |

### 5.1.1. COMPARISON OVER MDPS ON SEH

**Setup** We consider a task where the reward function is defined as a proxy model predicting binding affinity to the soluble epoxide hydrolase (sEH) target. We follow the experimental protocol of SynFlowNet. All models are post-trained for 5,000 steps, and for evaluation, 1,000 are randomly subsampled from generated molecules. We report the average sEH score, the positive ratio, the positive Top-100 average sEH, and diversity. Here, positive Top-100 is computed over the 100 highest-scoring synthesizable molecules among the generated samples. In addition, we report the Synthetic Accessibility (SA) score (Ertl & Schuffenhauer, 2009) as an external synthesizability metric and QED (Bickerton et al., 2012) as a measure of drug-likeness. We compare S3-GFN against representative methods utilizing different MDPs: the fragment-based **FragGFN** (Bengio et al., 2021) and the reaction-based **SynFlowNet** (**SFN**) (Cretu et al., 2025). Additionally, to isolate the efficacy of our contrastive learning objective, we include a reward shaping (**RTB+RS**) baseline. This method utilizes the same pre-trained SMILES prior but considers constraints directly via the task score, defining $r'(x) := r(x)\mathbb{I}[x \in \mathcal{X}']$. Further details are provided in Appendix B.2.

**Results** As shown in Figure 3 and Table 1, S3-GFN achieves a high positive ratio (0.945), indicating that the proposed replay training with contrastive auxiliary loss effectively concentrates probability mass within the synthesizable region, despite operating on a sequence-based MDP. Figure 4 illustrates the found synthetic pathways of our Top-2 positive molecules; the results from AiZynthFinder

*Table 2.* **Vina score** and **synthesizability** of Top-100 diverse modes on five receptors from LIT-PCBA. Results for baselines (†) are taken from Seo et al. (2025). Mean and standard deviations over three independent runs are reported, and best scores are shown in **bold**.

| | | Average Top-100 Vina Docking Score (kcal/mol, ↓) | | | | |
|---|---|---|---|---|---|---|
| Category | Method | ADRB2 | ALDH1 | ESR ago | ESR antago | FEN1 |
| Reaction | SynFlowNet† | -10.85 ± 0.10 | -10.69 ± 0.09 | -10.44 ± 0.05 | -10.27 ± 0.04 | -7.47 ± 0.02 |
| | RGFN† | -9.84 ± 0.21 | -9.93 ± 0.11 | -9.99 ± 0.11 | -9.72 ± 0.14 | -6.92 ± 0.06 |
| | RxnFlow† | -11.45 ± 0.05 | -11.26 ± 0.07 | -11.15 ± 0.02 | -10.77 ± 0.04 | -7.55 ± 0.02 |
| SMILES | S3-GFN | **-12.32** ± 0.026 | **-11.63** ± 0.015 | **-11.41** ± 0.035 | **-11.24** ± 0.011 | **-7.70** ± 0.011 |
| | | AiZynthFinder Success Rate (%, ↑) | | | | |
| Category | Method | ADRB2 | ALDH1 | ESR ago | ESR antago | FEN1 |
| Reaction | SynFlowNet† | 52.75 ± 1.09 | 57.00 ± 6.04 | 53.75 ± 9.52 | 56.50 ± 2.29 | 53.00 ± 8.92 |
| | RGFN† | 46.75 ± 6.86 | 47.50 ± 4.06 | 50.25 ± 2.17 | 49.25 ± 4.38 | 48.50 ± 6.58 |
| | RxnFlow† | 60.25 ± 3.77 | 63.25 ± 3.11 | 71.25 ± 4.15 | 66.50 ± 4.03 | 65.50 ± 4.09 |
| SMILES | S3-GFN | 100.0 ± 0.00 | 97.0 ± 2.94 | 96.67 ± 1.25 | 96.33 ± 1.24 | 99.0 ± 1.41 |

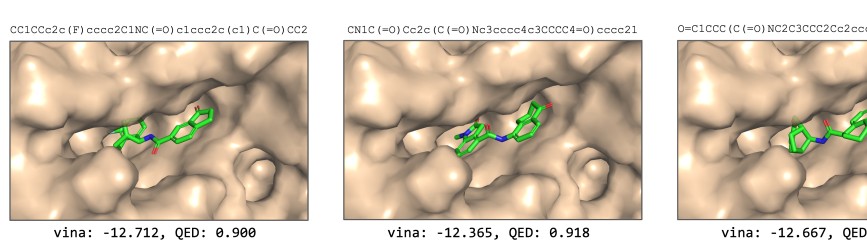

CC1CCc2c(F)cccc2C1NC(=O)c1ccc2c(c1)C(=O)CC2  CN1C(=O)Cc2c(C(=O)Nc3ccc4c3CCCC4=O)cccc21  O=C1CCC(C(=O)NC2C3CCC2Cc2ccccc2C3)c2ccccc21

vina: -12.712, QED: 0.900    vina: -12.365, QED: 0.918    vina: -12.667, QED: 0.879

*Figure 5.* **ALDH1 docking results (Top-3).** The molecules demonstrate shape complementarity with the target site, achieving strong predicted binding affinities (Vina scores $< -12.3$) and high drug-likeness (QED $> 0.87$) within the synthesizable space.

are provided in the Appendix C.3 Figure 9. Furthermore, S3-GFN attains higher Top-100 sEH scores among positive samples compared to SynFlowNet. This suggests that the proposed post-training reshapes the sampling distribution to favor higher-reward molecules while remaining within the synthesizable subspace. Importantly, favorable chemical properties likely inherited from the prior, such as high QED scores, are largely preserved after post-training. In Table 6, we analyze the reward-diversity trade-off, demonstrating that S3-GFN maintains competitive diversity while achieving higher rewards than baselines. While we outperform RTB+RS in Top-100 rewards, the margin is limited due to task saturation. We further analyze the distinct behaviors of reward shaping and the proposed auxiliary loss in Sections 5.2 and 5.3.

**Comparison with Saturn.** In Appendix C.2, we further compare S3-GFN with Saturn, a related SMILES-based method that uses retrosynthesis-driven reward shaping. Saturn differs from S3-GFN in both its optimization backbone, REINVENT (Olivecrona et al., 2017), and its pretrained prior.

### 5.1.2. STRUCTURED-BASED DRUG DISCOVERY

We further evaluate our method on pocket-specific optimization tasks in a structure-based drug discovery setting from

Seo et al. (2025). We perform docking-based optimization on five protein targets from LIT-PCBA (Tran-Nguyen et al., 2020) by using GPU-accelerated Uni-Dock (Yu et al., 2023) for Vina scores (Trott & Olson, 2010).

**Setup** Following RxnFlow (Seo et al., 2025), the reward is defined as a weighted combination of docking affinity and drug-likeness. After post-training, we generate 64,000 molecules and select a diverse set of Top-100 positive candidates. For these selected molecules, we report the average Vina docking score and the AiZynthFinder success rate. See Appendix B.3 for details.

**Results** As shown in Table 2, S3-GFN consistently achieves the lowest (better) average Vina docking scores across all five targets compared to reaction-based GFlowNet baselines. Interestingly, although reaction-based generation guarantees synthesizability within its own reaction template space by construction, its success rate under external AiZynthFinder evaluation drops to 72%. This indicates a gap between the reaction rules used during generation and those used during external validation: a molecule that is synthesizable under the generation-time template and building-block library may fail AiZynthFinder if it cannot find a route under its USPTO-based retrosynthesis model. In contrast, S3-GFN operates directly in the SMILES sequence space and regu-

*Table 3.* **Performance under constraint changes.** Results after adapting a model to a curated reaction set with additional Lipinski and BRENK constraints. We conduct 100 post-training steps under the changed constraints and evaluate for 1,000 samples.

|  | Zero-shot | RTB + RS | S3-GFN |
|---|---|---|---|
| Avg. sEH (↑) | 1.011 ± 0.001 | 0.943 ± 0.021 | **0.999** ± 0.006 |
| Positive Ratio (↑) | 0.740 ± 0.007 | **0.909** ± 0.019 | 0.883 ± 0.009 |
| Diversity (↑) | 0.761 ± 0.001 | 0.747 ± 0.006 | **0.758** ± 0.002 |
| Num. Unique (↑) | 927.0 ± 1.4 | 770.3 ± 17.0 | **929.0** ± 9.4 |
| Pos. Top100 sEH (↑) | 1.039 ± 0.001 | 1.037 ± 0.001 | **1.041** ± 0.001 |
| Pos. Top100 Div (↑) | 0.730 ± 0.008 | 0.707 ± 0.010 | **0.722** ± 0.006 |

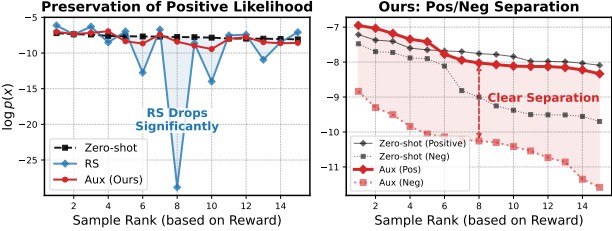

*Figure 6.* **Comparison of likelihood preservation and separation capability under constraint changes.** The contrastive auxiliary loss effectively separates positive and negative samples, while preserving $\log p(x)$ on Top-$K$ positive samples.

lates synthesizability through a soft constraint rather than a hard-coded generation process. This allows the model to bias sampling toward regions associated with synthesizable chemistry while retaining flexibility beyond a fixed rule set, as reflected by higher AiZynthFinder success rates.

We further provide the results on other protein targets from LIT-PCBA in Appendix C.5.

### 5.2. Analysis I: Realignment under Constraint Changes

In real-world molecular design, synthesizability is a context-dependent constraint rather than a fixed property. Because synthesis planning requires expert input, the effective constraint set often evolves during an iterative prototyping phase before synthesis. Constraints may be revised based on expert judgment, safety considerations, or immediate feedback on proposed candidates. This dynamic workflow requires models that can rapidly adapt to updated constraints within a small number of training iterations.

**Setup** Starting from a model trained under the original reaction set of 105 reactions, we update the constraints by (i) switching to a curated version of 32 reactions with a fewer synthesis steps, and (ii) adding chemical (e.g., Lipinski et al. (1997)) and structural (e.g., Brenk et al. (2008)) constraints; see Appendix B.4. We perform 100 steps of additional training and generate 1,000 molecules for evaluation. In Appendix C.6, we further evaluate more restrictive settings with only 15 and 10 available reactions.

*Table 4.* **AUC Top-10.** † means the results are taken from their original work. We report mean ± standard deviation with 3 runs.

|  | Method | GSK3$\beta$ | DRD2 |
|---|---|---|---|
| Reaction | Graph GA-ReaSyn† | 0.889 | 0.977 |
|  | SynGA† | 0.866 ± 0.072 | 0.976 ± 0.006 |
|  | SynFlowNet† | 0.691 ± 0.034 | 0.885 ± 0.027 |
| SMILES | REINVENT + RS | 0.830 ± 0.015 | 0.964 ± 0.006 |
|  | RTB + RS | 0.502 ± 0.042 | 0.783 ± 0.070 |
|  | S3-GFN | 0.807 ± 0.073 | 0.963 ± 0.004 |
|  | RTB + RS (Genetic Expl.) | 0.862 ± 0.048 | 0.961 ± 0.017 |
|  | S3-GFN (Genetic Expl.) | **0.905** ± 0.031 | **0.979** ± 0.005 |

**Results** Table 3 shows that both methods substantially increase the positive ratio under changed constraints compared to zero-shot evaluation, while S3-GFN preserves higher uniqueness and positive Top-100 sEH scores. This difference can be attributed to how constraint information is incorporated during post-training. Reward shaping redefines the reward and induces sustained suppression of negative samples throughout training. In SMILES generation, where positive and negative molecules often share long prefixes, this penalty can propagate to shared substructures, as observed in Figure 6. In contrast, the auxiliary loss operates on relative log-probabilities between positive and negative samples. Our gradient analysis in Appendix D shows that once positive and negative samples are sufficiently separated, the auxiliary gradients diminish. Therefore, the auxiliary loss encourages just enough likelihood reduction to distinguish infeasible continuations, without unnecessarily suppressing shared subtrajectories or positive samples.

### 5.3. Analysis II: Robustness in Sample-limited Regimes

We evaluate the robustness of our method in a sample-limited setting using the PMO benchmark (Gao et al., 2022a), where the oracle budget is limited to 10K.

**Setup** Following SynFlowNet, we evaluate performance on GSK3$\beta$ (Li et al., 2018) and DRD2 (Olivecrona et al., 2017). We report the area under the curve of Top-10 molecules (AUC Top-10) discovered during training. See Appendix B.5 for details. We include non-GFN baselines, GraphGA-ReaSyn (Lee et al., 2026) and SynGA (Lo et al., 2025). Additionally, we implement the method from Guo & Schwaller (2025) (denoted as 'REINVENT + RS'), using the same pre-trained prior for a fair comparison. Finally, we incorporate genetic exploration following the work by Kim et al. (2024a) to further leverage the off-policy nature of GFN, which allows the model to flexibly integrate external exploration strategies, a well-studied advantage of GFN (Kim et al., 2024b; Madan et al., 2025; Kim et al., 2025a).

**Results** Table 4 reports cumulative performance measured

by AUC Top-10. Interestingly, we observe a divergent response to penalty-based constraints between standard policy-gradient RL (REINVENT (Olivecrona et al., 2017)) and GFlowNets. While REINVENT + RS achieves strong performance (AUC 0.830 on GSK3$\beta$), applying the same reward-shaping strategy within a GFlowNet framework (RTB+RS) leads to a substantial degradation in performance (AUC 0.502). As illustrated in Figure 11 in Appendix C.7, the average reward of samples tend to converge early by failing to discovering new Top-100 candidates. Finally, when augmented with genetic exploration, S3-GFN attains the highest AUC among all evaluated GFN-based methods. As reported in Table 9, this trend extends to additional oracle tasks in the PMO benchmark, where our method achieves performance competitive with strong non-GFN baselines such as GraphGA-ReaSyn and SynGA. Although RTB+RS also benefits from off-policy data, a consistent performance gap remains, indicating that the proposed auxiliary loss more effectively incorporates high-reward trajectories discovered through external exploration. In Table 10, we study on efficacy of mutated negative samples on extended tasks.

## 6. Conclusion

We introduced S3-GFN, a post-training framework for synthesizable molecular generation that induces synthesizability through soft, distributional regularization rather than hard-constrained reaction-based MDP formulations. By leveraging rich chemical priors from pretrained SMILES language models and applying GFlowNet-based post-training, S3-GFN preserves the flexibility of sequence-based generation while suppressing negative region. The method fully utilize the off-policy nature of GFlowNets via replay-buffer management, and with the contrastive auxiliary loss, enabling explicit feasibility control without entangling it with reward optimization. Across multiple molecular generation tasks, S3-GFN achieves high synthesizability rates (up to 95%) while simultaneously improving task-specific rewards, demonstrating an effective balance between chemical feasibility and optimization performance.

**Limitation** While our method demonstrates promising results in simulation-based molecular design benchmarks, its validation is currently limited to in silico evaluations. Experimental verification through real-world chemical synthesis remains an important direction for future work.

## Impact Statement

The immediate goal of the method proposed here is to improve the synthesizability of molecules generated with machine learning. The ultimate goal is to facilitate the application of generative methods in drug discovery. While we are inspired by applications that could save human lives, such as antibiotics discovery, we acknowledge that our work can also have potential malicious applications, such as the development of chemical or biological weapons. We firmly stand against these unintended uses. Moreover, we acknowledge that if our contribution is meaningful, it is possible that the main benefit will be capitalized by powerful pharmaceutical companies, reducing the potential positive impact on people. Our hope and intention is that this work can contribute towards mitigating current and future pandemics, especially in underprivileged countries, which most acutely suffer global health challenges such as anti-microbial resistance.

## Acknowledgments

The authors thank Seonghwan Seo for valuable comments on this project. This work was supported by the Canada Biomedical Research Fund and Biosciences Research Infrastructure Fund (Grant CBRF2-2023-00107). This project was undertaken thanks to funding from IVADO and the Canada First Research Excellence Fund. This work was supported, in whole or in part, by the Gates Foundation INV-092957. The conclusions and opinions expressed in this work are those of the author(s) alone and shall not be attributed to the Foundation. The research was enabled in part by computational resources provided by the Digital Research Alliance of Canada (https://alliancecan.ca) and Mila (https://mila.quebec).

Hyeonah Kim is supported by the National Research Foundation of Korea (NRF) grant funded by the Korea government (MSIT) [RS-2025-00520529]. Minsu Kim acknowledges funding from the KAIST Jang Yeong Sil Fellow Program. Yoshua Bengio acknowledges support from the National Sciences and Engineering Council of Canada (NSERC), Samsung, and CIFAR. Yves V. Brun acknowledges support from the Canada 150 Research Chair in Bacterial Cell Biology.

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

## A. Use of Large Language Models

Large language models were used during manuscript preparation to improve grammar, clarity, and readability. All content was reviewed and edited by the authors.

## B. Experimental details

### B.1. S3-GFN

**Experimental Details on Molecular Generation Tasks**   Following the GFlowNet literature, we define the target sampling distribution using an energy-based form, with an inverse temperature parameter $\beta$, i.e., $p(x) \propto \exp(-\beta \mathcal{E}(x))$. Here, $\beta$ modulates the sharpness of the target distribution. This exponentiation strategy is widely adopted in the GFlowNet literature. In our main experiments on sEH and structure-based drug discovery, we employ $\beta = 25$. For the sample-efficient molecule generation task, we adopt $\beta = 50$, following the work by Kim et al. (2024a). In Appendix C.3, we investigate how lowering $\beta$ impacts the trade-off between diversity and rewards in the sEH task. We maintain separate replay buffers for positive and negative samples. In the positive buffer, we apply a reward-based eviction rule: when the buffer reaches capacity, a new candidate replaces the lowest-reward sample if it has higher reward and is not redundant. In contrast, the negative buffer follows a FIFO policy, where the oldest sample is removed upon insertion. The replay buffer size is set to 6,400 for the main experiments, following Seo et al. (2025), and 1,024 for the sample-limited benchmark, following Kim et al. (2024a). Across all molecular generation tasks, the minibatch size is fixed to $B = 64$.

**Implementation details of genetic exploration**   We employ genetic exploration only in the sample-limited setting described in Section 5.3, following the genetic-guided GFlowNet framework of Kim et al. (2024a). Our implementation largely matches the original method, with the key difference that samples generated via genetic operators are incorporated into training using our positive/negative replay-buffer rules. In particular, candidates produced by genetic exploration are classified by synthesizability and inserted into the corresponding buffer. Because genetic search yields approximately 20% of positive samples, we increase the number of offspring per genetic operation to 32 from 8 in the original implementation.

**Implementation details of mutated negative samples**   In molecular generation, unsynthesizable (negative) samples can be efficiently obtained by locally perturbing synthesizable (positive) molecules, whereas the reverse operation is generally non-trivial. We generate such negatives using the mutation operators introduced by Jensen (2019), which produce local structural variants while maintaining chemical validity. The mutation set includes atom and bond deletions, atom insertions, bond order changes, ring addition or deletion, and atom type substitutions. These operations ensure that negative samples remain close to their positive counterparts in chemical space, providing informative local contrast while avoiding artificially invalid structures. In addition, via this strategy, we are able to keep generating new negative samples even though our posterior mostly generate positive samples as trained.

### B.2. sEH benchmark task

In this benchmark, the target is the soluble epoxide hydrolase (sEH), a well-studied protein involved in cardiovascular and respiratory diseases (Imig & Hammock, 2009).

**Task setup**   Rewards are computed using a pretrained proxy model from Bengio et al. (2021), which predicts a normalized negative binding energy for a given molecule. During proxy training, raw binding energies are transformed to obtain strictly positive scores. The GFlowNet reward for the sEH task is obtained by further rescaling the transformed proxy output so that most values lie in the range $[0, 1]$.

**SynFlowNet training**   SynFlowNet is trained on synthetic pathway as trajectories starting from purchasable compounds, which are fragmentally built up using known reaction templates to form molecules with desired properties. The small molecular fragments were drawn from the commercially available Enamine building block library, and the reaction templates consist of 13 unimolecular and 92 bimolecular reactions obtained from two publicly available template libraries (Button et al., 2019; Hartenfeller et al., 2012).

Due to the sampling strategy employed by GFlowNet, which uses trajectory balance across synthetic pathway trajectories to promote exploration of multiple high-probability trajectories, the model generates a more diverse set of molecules. Since

training was restricted to an action space composed of chemically valid reactions from known templates, it ensures the synthesizability of the generated molecules.

For training, we used the original SynFlowNet implementation (Cretu et al., 2025) with the hyperparameters reported in the original paper in order to match the experimental setup used for the sEH task.[3] In particular, training was run for 5,000 steps, using a backward policy trained with maximum likelihood and Morgan fingerprint embeddings for the action space. The action space included 10,000 building blocks randomly sampled from the Enamine REAL database and all 105 reaction SMARTS templates provided in the original SynFlowNet implementation. The maximum trajectory length was set to 3.

**Fragment-based GFN (FragGFN) training**   In fragment-based GFlowNet formulations, molecules are constructed by sequentially assembling molecular fragments. Each fragment can have one or several possible attachment points. The discrete action space corresponds to jointly choosing an attachment site and a fragment to attach, with an additional stop action to terminate molecule construction. We use the publicly available implementation,[4] in which the fragment vocabulary consists of 105 distinct fragments extracted from the ZINC15 library.

For training, we directly use the hyperparameters provided in the code, but allowing replay training (64 samples from replay buffer). We also train the model with 5,000 steps with 64 batch size.

**RTB + Reward Shaping (RS)**   For a fair comparison, we apply reward shaping on top of our implementation and use the same hyperparameters, including replay training, except for the auxiliary loss. We use the same prior model, GP-MolFormer, and train the model using RTB in Equation (2) with modified reward $R'(x) = R(x)\mathbb{I}[x \in \mathcal{X}']$,

$$\mathcal{L}_{\text{RTB}}(\tau) = \left(\log Z_\theta + \log P_F^{\text{post}}(\tau; \theta) - \log R'(x) - \log P_F^{\text{prior}}(\tau)\right)^2.$$

**Evaluation**   For evaluation, we generate $64{,}000$ molecules and randomly subsample $\mathcal{S}$, with size of $1{,}000$ by following Cretu et al. (2025). Let $r(x)$ be the task-specific property score and $\mathcal{X}' \subset \mathcal{X}$ be the synthesizable molecular space defined by the retrosynthesis-based evaluator. We report the average task score,

$$\text{AvgScore}(\mathcal{S}) = \frac{1}{|\mathcal{S}|} \sum_{x \in \mathcal{S}} r(x), \tag{6}$$

and the positive ratio,

$$\text{PositiveRatio}(\mathcal{S}) = \frac{|\mathcal{S} \cap \mathcal{X}'|}{|\mathcal{S}|}. \tag{7}$$

We also report the positive Top-$K$ score, computed after canonicalizing and deduplicating generated molecules:

$$\text{PosTop}K(\mathcal{S}) = \frac{1}{K} \sum_{x \in \text{Top}_K(\mathcal{S} \cap \mathcal{X}')} r(x), \tag{8}$$

where $\text{Top}_K(\mathcal{S} \cap \mathcal{X}')$ denotes the $K$ highest-scoring unique synthesizable molecules among generated samples. We use $K = 100$ in the main experiments. In addition, we report molecular diversity over valid generated molecules. Following the SynFlowNet evaluation protocol, diversity is computed as the average pairwise Tanimoto distance between molecular fingerprints:

$$\text{Diversity}(\mathcal{S}) = \frac{2}{|\mathcal{S}|(|\mathcal{S}| - 1)} \sum_{i<j} \left(1 - \text{Tanimoto}\left(f(x_i), f(x_j)\right)\right), \tag{9}$$

where $f(x)$ denotes the molecular fingerprint of $x$.

### B.3. Structure-based drug discovery

For structure-based drug design, we follow the pocket-specific optimization in the work of Seo et al. (2025). We conduct experiments on receptors from the LIT-PCBA (Tran-Nguyen et al., 2020). We evaluate vina scoring with GPU-accelerated

---

[3]https://github.com/mirunacrt/synflownet
[4]https://github.com/recursionpharma/gflownet

UniDock (Yu et al., 2023). The reward is set as $R(x) = 0.5\widehat{\text{Vina}} + 0.5\text{QED}$, where $\widehat{\text{Vina}}$ is the normalized score as $\widehat{\text{Vina}}(x) = -0.1 \max(0, \text{Vina}(x))$.

For evaluation, diverse Top-100 molecules were selected based on the docking score after generating 64,000 candidates while ensuring a property constraint of QED $> 0.5$ and satisfying our training synthesizability (i.e., positive). Diversity is maintained using a pairwise Tanimoto dissimilarity filter of $0.5$ between the molecules. We excluded negative samples to make sure that the overall evaluations are conducted on synthesizable molecules.

### B.4. Fast adaptation under constraint changes

We study a setting in which synthesizability constraints are revised after a model has already been trained. We start from a single pretrained model trained under an initial reaction inventory consisting of 105 reaction templates with three synthesis steps. We then update the constraints along two dimensions. First, we replace the original reaction set with a curated subset of 32 widely used reaction templates (see below) with maximum two synthesis steps, favoring more conservative synthesis routes with fewer reaction steps. Second, we introduce additional molecule-level feasibility filters reflecting practical chemical and structural considerations. We use Lipinski rules (Lipinski et al., 1997)) as chemical property constraints and BRENK catalog (Brenk et al., 2008) as structural constraints. Here, since the proposed method is constraint-agnostic unless we can evaluate whether our molecule satisfies the given constraints, we can directly integrate those new constraints. Together, these changes define a more restrictive feasible region of the molecular space.

Importantly, during adaptation to the updated constraints, we reuse the existing replay buffer collected under the original constraints without additional evaluation. We reclassify stored molecules as feasible or infeasible according to the updated criteria, and then perform replay-only post-training updates for a fixed budget of 100 steps. Both RS-based and auxiliary-based methods start from the same pretrained model and are adapted using the same replay buffer, ensuring a controlled comparison that isolates the effect of the constraint-handling mechanism. Finally, we evaluate the adapted models by sampling 1,000 molecules from each resulting distribution and analyzing how probability mass is redistributed between molecules that remain feasible and those that become infeasible under the new constraints.

**Reaction curation**   The reaction SMARTS were originally taken from SynFormer (Gao et al., 2024) (92 bimolecular and 13 unimolecular reactions), which are also used in various reaction-based generation works (Cretu et al., 2025; Seo et al., 2025; Lo et al., 2025). We curate reactions based on domain knowledge to ensure that the retained templates correspond to one-step transformations that are routinely used in medicinal chemistry laboratories. Rare, overly specialized, or unusually challenging templates were removed, while a small number of missing but commonly used reactions were added based on expert input. We also exclude cross coupling reaction using aryl chlorides that are usually not as general as their bromide or iodide equivalent. The final curated set consists of 32 reactions (24 bimolecular and 8 unimolecular) that are broadly applicable in medicinal chemistry. The curated reaction set includes variations around amide coupling (including sulfonamides), urea formation from isocyanates, $S_N2$ functional group interconversions (e.g., halogen, primary alcohol, and nitrile), hydrolysis of acid chlorides, transesterification, $\alpha$-halogenation of carboxylic acids, aromatic nucleophilic substitution, azole cyclizations (imidazole, thiazole, Huisgen cycloaddition, and oxadiazole formation), Suzuki and Sonogashira couplings, as well as oxidation and reductive amination.

### B.5. Sample-efficient benchmark

For the experiments in the sample-limited setting in Section 5.3, we follow the official sample efficient molecule generation benchmark: the Practical Molecular Optimization (PMO) (Gao et al., 2022a). The PMO benchmark establishes a unified protocol for maximizing objective properties over chemical space. It comprises 23 diverse oracle functions derived from GuacaMol (Brown et al., 2019) and Therapeutics Data Commons (Huang et al., 2021), including bioactivity predictors (e.g., DRD2, GSK3$\beta$) and drug-likeness measures (e.g., QED), all normalized to $[0, 1]$. We omit VE To rigorously evaluate sample efficiency, the optimization budget is strictly limited to 10,000 oracle calls. Performance is primarily assessed via the Area Under the Curve (AUC) of the Top-10 scores. Notably, this benchmark apply early termination rules wherein the process terminates if Top-100 molecules remained same (fail to discover new Top-100 molecules) in few iteration. In this experiment, we follow the most of setup from (Kim et al., 2024a) except for the genetic exploration (see Appendix B.1).

**REINVENT + RS**   Based on the work from Guo & Schwaller (2025) and Guo et al. (2025), we implement reward shaping with REINVENT (Olivecrona et al., 2017) on top of S3-GFN implementation using the same prior model. Originally,

REINVENT also use replay buffer, so we use the same replay buffer following the original training process. The REINVENT loss is defined as

$$L(\theta) = \left[ \log p_\theta(x) - \left( \log p^{\text{prior}}(x) + \sigma R(x) \right) \right]^2.$$

We follow the implementation of the PMO benchmark.[5] The reward $R(x)$ is replaced by $R'(x) = R(x)\mathbb{I}[x \in \mathcal{X}']$.

---

[5]https://github.com/wenhao-gao/mol_opt

# C. Further experimental results

## C.1. Studies on the auxiliary coefficient $\alpha$

In this section, we analyze the impact of the penalty coefficient $\alpha$ on exploration dynamics. As an extension of Figure 2, Figure 7 shows that as $\alpha$ increases, the contrastive auxiliary loss suppresses negative samples more aggressively, resulting in a more conservative policy. Conversely, reducing $\alpha$ relaxes this penalty, making the model generate negative samples near the constraint boundary. However, in general, S3-GFN demonstrates robustness in balancing constraint satisfaction and reward distribution learning within the positive area even with changes in $\alpha$.

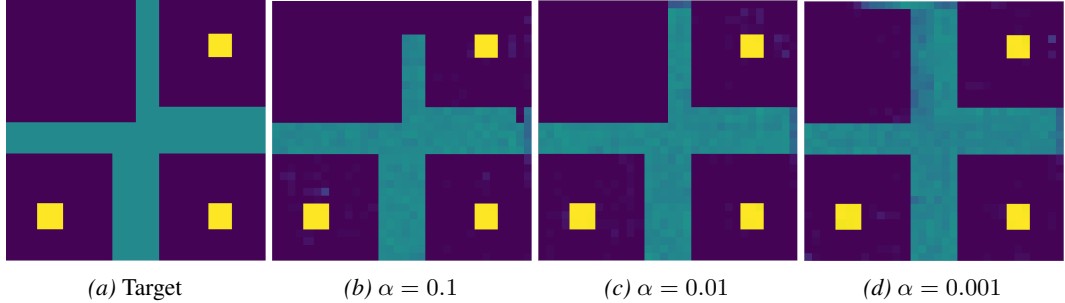

*(a)* Target        *(b)* $\alpha = 0.1$        *(c)* $\alpha = 0.01$        *(d)* $\alpha = 0.001$

*Figure 7.* Behaviors with a different aux coefficient $\alpha$.

## C.2. Comparison with a SMILES-based method

We additionally compare S3-GFN with Saturn (Guo & Schwaller, 2025; Guo et al., 2025), a SMILES-based molecular generation framework that incorporates retrosynthesis feedback through reward shaping. We evaluate Saturn on the sEH task, one of the main molecular generation benchmarks considered in this paper, and on the sample-efficient benchmark. Note that Saturn is originallyh proposed under the sample-efficient regime. We use the original Saturn codebase[6] with only minimal modifications required to adapt it to our task.

To ensure a fair comparison, we use the same synthesizability evaluator as in our experiments. The original Saturn implementation uses Syntheseus (Maziarz et al., 2025) for retrosynthesis evaluation, but we found it computationally expensive for our setting. More importantly, using the same evaluator ensures that both methods are compared under the same synthesizable set $\mathcal{X}'$.

**Results on sEH.** Table 5 reports the comparison on the sEH task. Compared to Saturn, S3-GFN achieves a higher positive ratio, higher average sEH score, higher positive Top-100 sEH score, and substantially higher diversity. These results suggest that, while Saturn effectively improves synthesizability through retrosynthesis-based reward shaping, S3-GFN better preserves both high-reward optimization and diversity within the synthesizable region.

*Table 5.* **Result of SMILES-based models on sEH.** Mean and standard deviations are reported with three independent runs.

| Method | Positive (↑) | Pos. Top100 sEH (↑) | sEH (↑) | Diversity (↑) |
|--------|--------------|---------------------|---------|---------------|
| Saturn | $0.885 \pm 0.020$ | $1.027 \pm 0.002$ | $0.906 \pm 0.018$ | $0.390 \pm 0.136$ |
| S3-GFN | $0.945 \pm 0.009$ | $1.043 \pm 0.001$ | $1.009 \pm 0.000$ | $0.764 \pm 0.000$ |

We also include Saturn in the sample-efficient PMO benchmark comparison in Table 4, together with the other baselines. Saturn performs strongly on GSK3$\beta$ and DRD2, while S3-GFN with genetic exploration achieves the best aggregate performance over all 23 tasks.

Overall, both Saturn and S3-GFN benefit from strong SMILES-based representations. However, S3-GFN achieves stronger performance on the main benchmarks considered here. These results are consistent with the role of off-policy mechanisms in S3-GFN: positive and negative replay buffers improve synthesizability. Furthermore, genetic off-policy exploration improves reward optimization in the sample-efficient benchmark.

---

[6]https://github.com/schwallergroup/saturn

## C.3. Additional results on sEH

Table 6 shows additional comparisons of the model performance on sEH target as an extension of the results in Table 1. The synthetic accessibility score (SA), AiZynthFinder success rate (AiZynth), drug-likeness score (QED), and the molecular weight (Mol. weight) of the generated molecules of S3-GFN further supported the model, consistent with our conclusions.

*Table 6.* **Comparison on sEH.** Mean ± standard deviation over 3 runs are reported. Best scores among synthesizability-aware approaches are shown in **bold**. Full results of Table 1

| Category | Method | Positive (↑) | Pos. Top100 sEH (↑) | sEH (↑) | Diversity (↑) | SA (↓) | AiZynth (↑) | QED (↑) | Mol. weight (↓) |
|---|---|---|---|---|---|---|---|---|---|
| Fragment | FragGFN | 0.0 ± 0.000 | 0.969 ± 0.003 | 0.790 ± 0.002 | 0.822 ± 0.002 | 4.806 ± 1.000 | 0.0 ± 0.000 | 0.189 ± 0.010 | 688.20 ± 15.00 |
| Reaction | SynFlowNet | **1.0** | 0.989 ± 0.005 | 0.907 ± 0.007 | 0.801 ± 0.022 | 2.876 ± 0.150 | 0.727 ± 0.255 | 0.694 ± 0.054 | 356.00 ± 6.31 |
| SMILES | Prior | 0.299 ± 0.014 | 0.628 ± 0.008 | 0.482 ± 0.003 | 0.877 ± 0.001 | 3.035 ± 0.022 | 0.730 ± 0.019 | 0.685 ± 0.007 | 368.98 ± 2.61 |
| | S3-GFN ($\beta = 15$) | 0.928 ± 0.001 | 1.021 ± 0.000 | 0.919 ± 0.003 | **0.803 ± 0.002** | 2.427 ± 0.027 | 0.983 ± 0.005 | **0.748 ± 0.004** | **340.41 ± 2.77** |
| | S3-GFN ($\beta = 25$) | 0.945 ± 0.009 | **1.043 ± 0.001** | **1.009 ± 0.000** | 0.764 ± 0.000 | **2.364 ± 0.016** | **0.990 ± 0.008** | 0.696 ± 0.007 | 350.35 ± 2.84 |

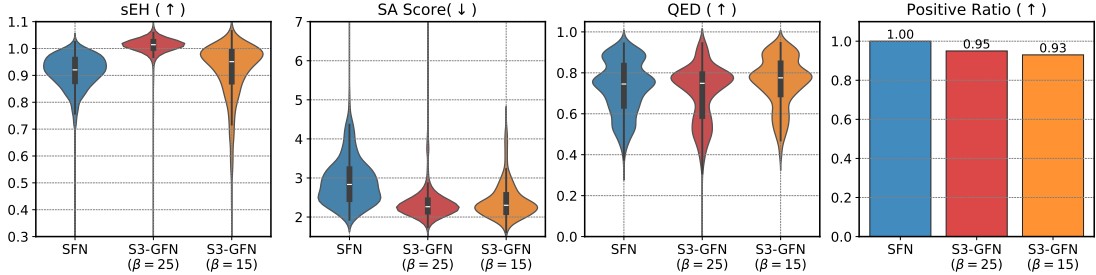

*Figure 8.* **Comparisons over $\beta$.** S3-GFN with $\beta = 15$ tends to more diverse compared to S3-GFN with $\beta = 25$.

We also provide examples of retrosynthesis results from AiZynthFinder for Top-2 candidates in Figure 9. These results show that molecules generated using S3-GFN have high chances of having valid synthesis routes under given conditions (with given 105 reaction templates and Enamine Stock building blocks) and AiZynthFinder.

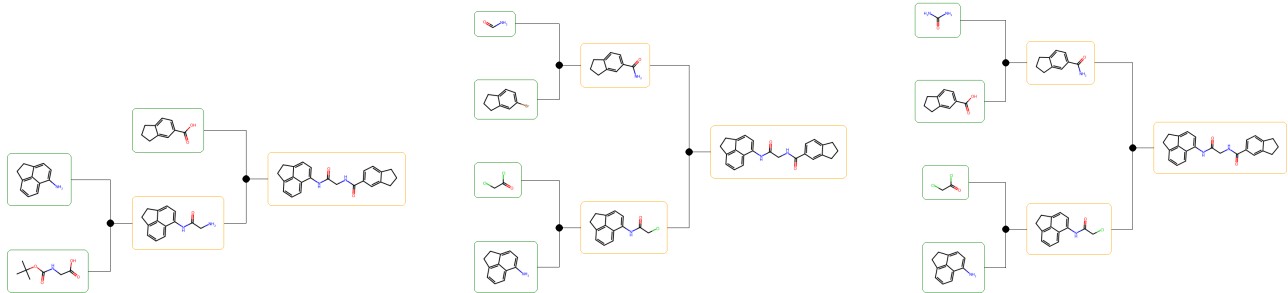

*(a)* **Different synthesis route for** `O=C(CNC(=O)c1ccc2c(c1)CCC2)Nc1ccc2c3c(cccc13)CC2` (sEH: 1.074)

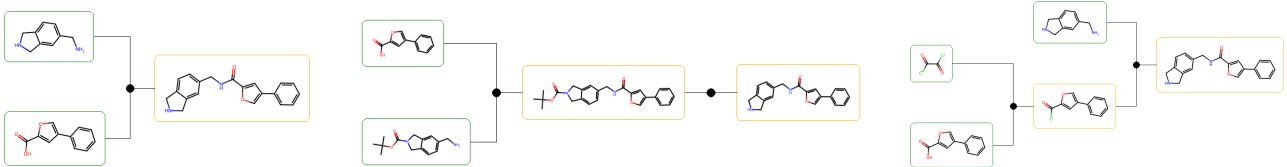

*(b)* **Different synthesis route for** `O=C(NCc1ccc2c(c1)CNC2)c1cc(-c2ccccc2)co1` (sEH: 1.063)

*Figure 9.* AiZynthFinder results on Top-2 candidates in Figure 4.

## C.4. Generated Top-1 candidates on SBDD

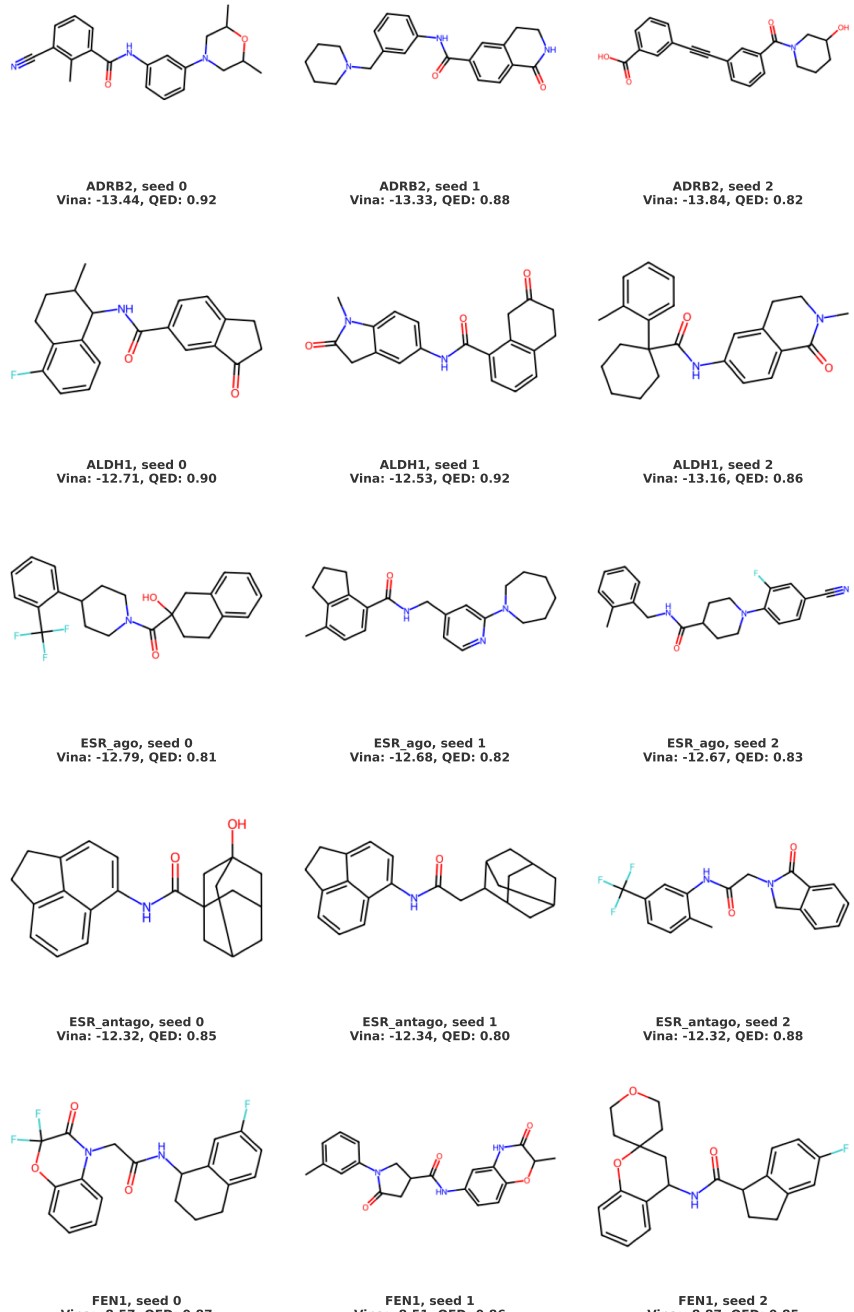

*Figure 10.* **Examples of generated positive molecules by S3-GFN.** We visualize the best candidate based on the vina score over all seeds.

## C.5. Additional results on LIT-PCBA

We provide additional results with 3 independent runs on the targets from LIT-PCBA as follows.

*Table 7.* **Vina score** of diverse Top-100 modes on LIT-PCBA. Mean $\pm$ standard deviations over 3 runs are reported.

|          | Method               | GBA              | IDH1              | KAT2A             | MAPK1            | MTORC1            |
| -------- | -------------------- | ---------------- | ----------------- | ----------------- | --------------- | ----------------- |
| Reaction | SynFlowNet[†]        | -9.27 $\pm$ 0.06 | -10.40 $\pm$ 0.08 | -9.41 $\pm$ 0.04  | -8.92 $\pm$ 0.05 | -10.84 $\pm$ 0.03 |
|          | RGFN[†]              | -8.48 $\pm$ 0.06 | -9.49 $\pm$ 0.13  | -8.53 $\pm$ 0.11  | -8.22 $\pm$ 0.15 | -9.89 $\pm$ 0.06  |
|          | RxnFlow[†]           | -9.62 $\pm$ 0.04 | -10.95 $\pm$ 0.05 | -9.73 $\pm$ 0.03  | -9.30 $\pm$ 0.01 | -11.39 $\pm$ 0.09 |
| SMILES   | S3-GFN               | **-9.76** $\pm$ 0.08 | **-11.43** $\pm$ 0.03 | **-10.02** $\pm$ 0.10 | **-9.38** $\pm$ 0.03 | **-11.73** $\pm$ 0.01 |

|          | Method               | OPRK1             | PKM2              | PPARG            | TP53             | VDR               |
| -------- | -------------------- | ----------------- | ----------------- | ---------------- | ---------------- | ----------------- |
| Reaction | SynFlowNet[†]        | -10.34 $\pm$ 0.07 | -11.98 $\pm$ 0.12 | -9.40 $\pm$ 0.05 | -7.90 $\pm$ 0.10 | -11.62 $\pm$ 0.13 |
|          | RGFN[†]              | -9.61 $\pm$ 0.11  | -10.96 $\pm$ 0.18 | -8.53 $\pm$ 0.07 | -7.07 $\pm$ 0.06 | -10.86 $\pm$ 0.11 |
|          | RxnFlow[†]           | -10.84 $\pm$ 0.03 | -12.53 $\pm$ 0.02 | -9.73 $\pm$ 0.02 | -8.09 $\pm$ 0.06 | -12.30 $\pm$ 0.07 |
| SMILES   | S3-GFN               | **-11.48** $\pm$ 0.01 | **-13.15** $\pm$ 0.03 | **-9.89** $\pm$ 0.02 | **-8.33** $\pm$ 0.10 | **-12.86** $\pm$ 0.10 |

## C.6. Realignment under stricter synthesizability constraints

To further examine adaptation under more restrictive constraints, we consider harsher constraint changes by reducing the allowed reaction subset from 32 reactions to 15 and 10 reactions. These settings substantially lower the zero-shot positive ratio, making realignment more challenging.

*Table 8.* **Performance under harsher constraint changes.** Mean ± standard deviation over 3 runs are reported.

| | $|\mathcal{R}| = 15$ # positives in buffer: 2241 | | | $|\mathcal{R}| = 10$ # positives in buffer: 1073 | | |
| --- | --- | --- | --- | --- | --- | --- |
| | Zero-shot | RTB + RS | S3-GFN | Zero-shot | RTB + RS | S3-GFN |
| Avg. sEH (↑) | **1.010 ± 0.001** | 0.943 ± 0.001 | 1.007 ± 0.001 | **1.010 ± 0.001** | 0.862 ± 0.042 | 1.007 ± 0.002 |
| Positive Ratio (↑) | 0.620 ± 0.023 | **0.865 ± 0.020** | 0.838 ± 0.012 | 0.174 ± 0.018 | 0.637 ± 0.061 | **0.844 ± 0.010** |
| Num. Unique (↑) | **923.7 ± 3.1** | 795.0 ± 8.5 | 861.3 ± 13.7 | **923.7 ± 3.1** | 550.7 ± 22.4 | 687.3 ± 14.4 |
| Pos. Top100 sEH (↑) | 1.035 ± 0.002 | 1.029 ± 0.001 | **1.037 ± 0.000** | 1.035 ± 0.002 | 1.016 ± 0.004 | **1.038 ± 0.000** |

## C.7. Sample-efficient benchmark

Figure 11 shows the performance of the different models on GSK3$\beta$ and DRD2 tasks.

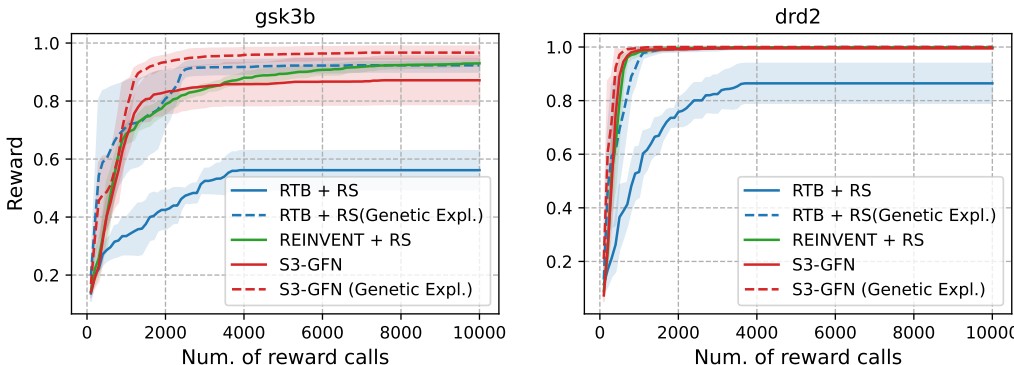

*Figure 11.* **Top-10 optimization curve on GSK3$\beta$ and DRD2.** Average reward of Top-10 molecules generated by different models.

We extend the analysis to other oracles in the PMO benchmark as shown in Table 9. The area under the curve (AUC) of top-10 generated molecules during the optimization of S3-GFN are compared to baseline models GraphGA-ReaSyn (Lee et al., 2026) and SynGA (Lo et al., 2025) in addition to Table 4. The results show that even though GraphGA-ReaSyn and SynGA have the best performance in some of the tasks, S3-GFN has the best overall performance which supports our conclusions.

*Table 9.* **AUC Top-10 for different Oracles.** We provide full results of Table 4. † means the results are taken from their original work. Best scores are shown in **bold**.

| Oracle | Graph GA ReaSyn[†] | SynGA[†] | Saturn | S3-GFN | S3-GFN (Genetic Expl.) |
|---|---|---|---|---|---|
| Albu. Sim. | - | $0.649 \pm 0.058$ | $0.656 \pm 0.046$ | $0.603 \pm 0.016$ | $\mathbf{0.793 \pm 0.007}$ |
| Amlo. MPO | **0.678** | $0.573 \pm 0.019$ | $0.529 \pm 0.009$ | $0.552 \pm 0.005$ | $0.580 \pm 0.010$ |
| Cele. Redisc. | **0.754** | $0.494 \pm 0.063$ | $0.487 \pm 0.020$ | $0.465 \pm 0.009$ | $0.549 \pm 0.046$ |
| Deco Hop | - | $0.629 \pm 0.014$ | $0.684 \pm 0.012$ | $0.597 \pm 0.001$ | $\mathbf{0.780 \pm 0.081}$ |
| DRD2 | 0.973 | $0.976 \pm 0.006$ | $0.978 \pm 0.002$ | $0.963 \pm 0.004$ | $\mathbf{0.979 \pm 0.005}$ |
| Fexo. MPO | **0.788** | $0.773 \pm 0.018$ | $0.707 \pm 0.006$ | $0.702 \pm 0.003$ | $0.744 \pm 0.008$ |
| GSK3$\beta$ | 0.851 | $0.866 \pm 0.072$ | $\mathbf{0.917 \pm 0.007}$ | $0.807 \pm 0.073$ | $0.905 \pm 0.031$ |
| Isom. C7H8. | - | $0.840 \pm 0.016$ | $\mathbf{0.972 \pm 0.005}$ | $0.911 \pm 0.035$ | $0.959 \pm 0.008$ |
| Isom. C9H10. | - | $0.707 \pm 0.040$ | $\mathbf{0.828 \pm 0.007}$ | $0.812 \pm 0.011$ | $0.809 \pm 0.012$ |
| JNK3 | 0.741 | $0.683 \pm 0.132$ | $\mathbf{0.865 \pm 0.053}$ | $0.824 \pm 0.018$ | $0.797 \pm 0.027$ |
| Median 1 | 0.293 | $0.254 \pm 0.017$ | $\mathbf{0.310 \pm 0.022}$ | $0.245 \pm 0.015$ | $0.285 \pm 0.002$ |
| Median 2 | 0.259 | $0.226 \pm 0.009$ | $0.242 \pm 0.004$ | $0.215 \pm 0.001$ | $\mathbf{0.262 \pm 0.015}$ |
| Mest. Sim. | - | $\mathbf{0.480} \pm 0.008$ | $0.376 \pm 0.009$ | $0.394 \pm 0.001$ | $0.451 \pm 0.013$ |
| Osim. MPO | 0.820 | $0.820 \pm 0.003$ | $0.804 \pm 0.006$ | $0.801 \pm 0.003$ | $\mathbf{0.827 \pm 0.005}$ |
| Peri. MPO | **0.560** | $0.556 \pm 0.032$ | $0.462 \pm 0.005$ | $0.480 \pm 0.005$ | $0.513 \pm 0.012$ |
| QED | - | $0.938 \pm 0.001$ | $0.941 \pm 0.000$ | $0.941 \pm 0.000$ | $\mathbf{0.942 \pm 0.000}$ |
| Rano. MPO | 0.742 | $\mathbf{0.802 \pm 0.009}$ | $0.675 \pm 0.030$ | $0.223 \pm 0.315$ | $0.754 \pm 0.021$ |
| Scaffold Hop | - | $0.532 \pm 0.014$ | $0.499 \pm 0.026$ | $0.485 \pm 0.003$ | $\mathbf{0.561 \pm 0.028}$ |
| Sita. MPO | 0.342 | $0.348 \pm 0.022$ | $0.426 \pm 0.016$ | $0.444 \pm 0.026$ | $\mathbf{0.451 \pm 0.057}$ |
| Thio. Redisc. | - | $0.433 \pm 0.033$ | $0.436 \pm 0.011$ | $0.431 \pm 0.009$ | $\mathbf{0.503 \pm 0.018}$ |
| Trog. Redisc. | - | $\mathbf{0.322 \pm 0.013}$ | $0.276 \pm 0.004$ | $0.297 \pm 0.018$ | $0.304 \pm 0.005$ |
| Zale. MPO | 0.492 | $0.465 \pm 0.017$ | $0.478 \pm 0.007$ | $0.494 \pm 0.010$ | $\mathbf{0.507 \pm 0.007}$ |
| Sum | - | 13.366 | 13.548 | 12.686 | **14.255** |

Additionally, we explored the performance of S3-GFN when trained with mutated negative samples across different oracles

as shown in Table 10. The implementation of mutation is described in detail in Section B.1.

*Table 10.* **Comparing S3-GFN with and without mutated negative samples.** Mean and standard deviation of AUC Top-10 are reported over different Oracles. † means the results are taken from their original work. Best scores are shown in **bold**.

| Oracle | w/ $x_{\mathrm{mut}}^-$ | w/o $x_{\mathrm{mut}}^-$ |
|---|---|---|
| Albu. Sim. | $0.793 \pm 0.007$ | $\mathbf{0.844 \pm 0.032}$ |
| Amlo. MPO | $\mathbf{0.580 \pm 0.010}$ | $0.567 \pm 0.002$ |
| Cele. Redisc. | $\mathbf{0.549 \pm 0.046}$ | $0.523 \pm 0.015$ |
| Deco Hop | $\mathbf{0.780 \pm 0.081}$ | $0.757 \pm 0.094$ |
| DRD2 | $\mathbf{0.979 \pm 0.005}$ | $0.976 \pm 0.003$ |
| Fexo. MPO | $0.744 \pm 0.008$ | $\mathbf{0.745 \pm 0.010}$ |
| GSK3$\beta$ | $\mathbf{0.905 \pm 0.031}$ | $0.886 \pm 0.011$ |
| Isom. C7H8. | $\mathbf{0.959 \pm 0.008}$ | $0.954 \pm 0.016$ |
| Isom. C9H10. | $0.809 \pm 0.012$ | $\mathbf{0.844 \pm 0.018}$ |
| JNK3 | $0.797 \pm 0.027$ | $\mathbf{0.885 \pm 0.012}$ |
| Median 1 | $\mathbf{0.285 \pm 0.002}$ | $0.280 \pm 0.015$ |
| Median 2 | $\mathbf{0.262 \pm 0.015}$ | $0.258 \pm 0.004$ |
| Mest. Sim. | $0.451 \pm 0.013$ | $\mathbf{0.453 \pm 0.015}$ |
| Osim. MPO | $\mathbf{0.827 \pm 0.005}$ | $0.826 \pm 0.006$ |
| Peri. MPO | $0.513 \pm 0.012$ | $\mathbf{0.519 \pm 0.021}$ |
| QED | $0.942 \pm 0.000$ | $0.942 \pm 0.000$ |
| Rano. MPO | $\mathbf{0.754 \pm 0.012}$ | $0.723 \pm 0.026$ |
| Scaffold Hop | $0.561 \pm 0.028$ | $\mathbf{0.565 \pm 0.031}$ |
| Sita. MPO | $\mathbf{0.451 \pm 0.057}$ | $0.402 \pm 0.023$ |
| Thio. Redisc. | $\mathbf{0.503 \pm 0.018}$ | $0.490 \pm 0.016$ |
| Trog. Redisc. | $0.304 \pm 0.005$ | $\mathbf{0.308 \pm 0.003}$ |
| Zale. MPO | $0.507 \pm 0.007$ | $0.507 \pm 0.007$ |
| Sum | 14.255 | 14.254 |

## D. Gradient Analysis: Contrastive Replay vs. Reward-Shaped RTB

### D.1. Behavior Under Sufficient Score Separation

We analyze how the replay update behaves once positive and negative samples are already sufficiently separated in score space. Following the manuscript notation, let $D^+$ and $D^-$ denote the positive and negative replay buffers, and let $B^+ \sim D^+$ and $B^- \sim D^-$ denote the replay mini-batches used in replay training. We write

$$s_\theta(\tau) := \log P_F^{\text{post}}(\tau; \theta), \qquad s^{\text{prior}}(\tau) := \log P_F^{\text{prior}}(\tau).$$

**Replay objective.** The replay objective is

$$\mathcal{L}_{\text{replay}} = \mathcal{L}_{\text{RTB}}^+ + \alpha \mathcal{L}_{\text{aux}}, \qquad \mathcal{L}_{\text{RTB}}^+ = \frac{1}{|B^+|} \sum_{\tau \in B^+} \mathcal{L}_{\text{RTB}}(\tau), \tag{10}$$

where

$$\mathcal{L}_{\text{RTB}}(\tau) = \Big( \log Z_\theta + s_\theta(\tau) - \log R(x) - s^{\text{prior}}(\tau) \Big)^2. \tag{11}$$

**Gradient of the positive RTB term.** Define the RTB residual

$$\delta^+(\tau) := \log Z_\theta + s_\theta(\tau) - \log R(x) - s^{\text{prior}}\tau). \tag{12}$$

Then

$$\mathcal{L}_{\text{RTB}}(\tau) = \big(\delta^+(\tau)\big)^2.$$

Applying the chain rule,

$$\nabla_\theta \mathcal{L}_{\text{RTB}}(\tau) = 2\,\delta^+(\tau)\,\nabla_\theta \delta^+(\tau).$$

Since $R(x)$ and $s_\theta^{\text{prior}}(\tau)$ are fixed with respect to $\theta$,

$$\nabla_\theta \delta^+(\tau) = \nabla_\theta \log Z_\theta + \nabla_\theta s_\theta(\tau).$$

Therefore, the gradient of the positive RTB term is

$$\nabla_\theta \mathcal{L}_{\text{RTB}}^+ = \frac{1}{|B^+|} \sum_{\tau \in B^+} 2\,\delta^+(\tau)\Big(\nabla_\theta \log Z_\theta + \nabla_\theta s_\theta(\tau)\Big). \tag{13}$$

**Auxiliary loss and its gradient coefficients.** The auxiliary loss is

$$\mathcal{L}_{\text{aux}} = -\sum_{\tau_i^+ \in B^+} \log \frac{e^{s_\theta(\tau_i^+)}}{e^{s_\theta(\tau_i^+)} + \sum_{\tau_j^- \in B^-} e^{s_\theta(\tau_j^-)}}. \tag{14}$$

Define

$$S_- := \sum_{\tau_j^- \in B^-} e^{s_\theta(\tau_j^-)}.$$

Then (14) can be rewritten as

$$\mathcal{L}_{\text{aux}} = \sum_{\tau_i^+ \in B^+} \ell_i, \qquad \ell_i := \log\big(e^{s_\theta(\tau_i^+)} + S_-\big) - s_\theta(\tau_i^+). \tag{15}$$

For each positive sample $\tau_i^+$, the partial derivatives of $\ell_i$ with respect to the positive and negative scores are

$$\frac{\partial \ell_i}{\partial s_\theta(\tau_i^+)} = -\frac{S_-}{e^{s_\theta(\tau_i^+)} + S_-}, \qquad \frac{\partial \ell_i}{\partial s_\theta(\tau_j^-)} = \frac{e^{s_\theta(\tau_j^-)}}{e^{s_\theta(\tau_i^+)} + S_-}. \tag{16}$$

Now suppose that positive and negative samples are already sufficiently separated, in the sense that

$$e^{s_\theta(\tau_i^+)} \gg S_- \qquad \text{for each } \tau_i^+ \in B^+. \tag{17}$$

Under (17), both coefficients in (16) become small:

$$\frac{S_-}{e^{s_\theta(\tau_i^+)} + S_-} \approx 0, \qquad \frac{e^{s_\theta(\tau_j^-)}}{e^{s_\theta(\tau_i^+)} + S_-} \approx 0.$$

Hence,

$$\nabla_\theta \mathcal{L}_{\text{aux}} \approx 0. \tag{18}$$

Combining (13) and (18), the replay update becomes

$$\nabla_\theta \mathcal{L}_{\text{replay}} = \nabla_\theta \mathcal{L}_{\text{RTB}}^+ + \alpha \nabla_\theta \mathcal{L}_{\text{aux}} \approx \nabla_\theta \mathcal{L}_{\text{RTB}}^+. \tag{19}$$

That is, once sufficient separation is achieved, the auxiliary term contributes only a diminishing gradient, and the replay update is effectively dominated by the positive RTB term.

