# OpenReview forum: "Synthesizable Molecular Generation via Soft-constrained GFlowNets with Rich Chemical Priors"
_ICML.cc/2026/Conference — ICML 2026 regular_

### Official Review · Reviewer_npKo · 2026-03-10

**Soundness:** 3
**Presentation:** 3
**Significance:** 3
**Originality:** 2
**Overall Recommendation:** 4
**Confidence:** 4

**Summary:**

S3-GFN is a post-training framework that induces synthesizability in sequence-based SMILES generation via soft constraints, rather than hard-constrained reaction-based MDPs. The core idea is to initialize from a pretrained SMILES language model (GP-MolFormer) and apply GFlowNet post-training (RTB) with a contrastive auxiliary loss. Two separate replay buffers are maintained for synthesizable (positive) and unsynthesizable (negative) samples. A contrastive loss is applied exclusively during replay training to relatively suppress the log-probability of negative trajectories. Crucially, this decouples constraint enforcement from reward optimization, unlike naive reward shaping (RS), which entangles the two and can inadvertently penalize shared substructures between positive and negative molecules. The method also supports fast adaptation to changing constraints by simply reclassifying existing replay buffer contents with minimal additional training steps. Empirically, S3-GFN achieves over 95% synthesizability while outperforming reaction-based GFlowNet baselines on reward across multiple benchmarks (sEH, LIT-PCBA SBDD, PMO).

**Compliance With Llm Reviewing Policy:**

Affirmed.

**Final Justification:**

Thank you for the additional experiments comparing RTB against SQL and SAC under the same constraint formulation. The results provide useful empirical justification for the choice of trajectory-level matching in this sparse-reward setting, which largely addresses my concern in Q1. I will raise my score accordingly.

**Key Questions For Authors:**

Q1. In the SMILES sequence generation setting, the trajectory-to-terminal mapping is injective (i.e., $PB(\tau|x) = 1$), which means the TB/RTB objective reduces to Path Consistency Learning and the GFlowNet framework collapses to a form closely related to MaxEnt RL or standard autoregressive policy optimization. Given this, it is unclear what specific advantage GFlowNet post-training provides over simpler alternatives such as PPO/REINFORCE-style MaxEnt RL or direct autoregressive fine-tuning (e.g., supervised fine-tuning on high-reward synthesizable samples). The paper includes REINVENT+RS as a baseline, but REINVENT also uses reward shaping, conflating two differences at once. A cleaner ablation that isolates the effect of the GFlowNet training objective itself against MaxEnt RL or autoregressive fine-tuning under the same contrastive constraint formulation seems necessary to validate the core architectural choice. If the authors can demonstrate a clear empirical or theoretical advantage of GFlowNet over these simpler baselines under the proposed soft constraint scheme, it would substantially strengthen the justification for the method design.

Q2. The positive/negative classification relies on a retrosynthesis-based heuristic (using 105 reaction templates from SynFlowNet). This oracle is itself imperfect and template-limited. How sensitive is S3-GFN's performance to the choice of synthesizability oracle? In particular, does training with one oracle (template-based) and evaluating with another (AiZynthFinder/USPTO) introduce a distribution mismatch that artificially favors S3-GFN over reaction-based baselines, which are constrained to the same template set at both train and test time?

Q3. The auxiliary loss relies heavily on the quality and diversity of negative samples in D−. The paper generates negatives via local mutations of positive samples (Jensen 2019 operators). However, mutation-based negatives are structurally very close to positives, which may produce an overly easy contrastive signal. Have the authors studied how the diversity or difficulty of negative samples affects performance? Would harder negatives (e.g., from a separate generative model) improve constraint satisfaction further?

Q4. During fast adaptation experiments, the replay buffer collected under the original constraints is reused and reclassified under new constraints. However, if the new constraint space is substantially different, the existing buffer may be heavily imbalanced (e.g., very few positives remain). How does S3-GFN behave in more drastic constraint shifts, and is there a risk of the method degrading to reward shaping-like behavior when the positive buffer becomes sparse?

**Limitations:**

The authors acknowledge the key limitation that validation is entirely in silico and that experimental verification through real-world chemical synthesis remains future work. They also briefly discuss potential negative societal impacts, including misuse for chemical or biological weapons development. These disclosures are appreciated.
However, two additional limitations warrant discussion. First, the method's performance is inherently tied to the quality and coverage of the synthesizability oracle used during training; sensitivity to oracle choice is not discussed. Second, the scalability of the replay buffer approach under large constraint shifts or highly sparse positive regions is not characterized. Including these points would provide a more complete picture of the method's practical boundaries.

**Strengths And Weaknesses:**

Strengths
- The contrastive auxiliary loss provides a clean decoupling of constraint enforcement from reward optimization, avoiding the prefix penalty propagation issue inherent to reward shaping in SMILES generation
- Fast adaptation under evolving constraints via replay buffer reclassification is practically well-motivated and empirically demonstrated with minimal additional training steps
- Strong performance on AiZynthFinder despite training with a different synthesizability oracle suggests genuine generalization of the learned synthesizability bias
- The 2D grid world experiment provides a clean and interpretable proof-of-concept for the role of the auxiliary loss

Weaknesses
- In the injective SMILES setting ($PB(\tau|x) = 1$), the RTB objective reduces to a form closely related to MaxEnt RL or autoregressive fine-tuning, yet no such baselines are included. The only RL baseline (REINVENT+RS) conflates the training objective with reward shaping, making it impossible to attribute performance gains to the GFlowNet framework itself. This is a fundamental gap in the experimental validation that must be addressed before the method's design choices can be justified.
- The synthesizability oracle used during training (105 reaction templates) differs from the evaluation oracle (AiZynthFinder/USPTO), which may systematically favor S3-GFN over reaction-based baselines that are constrained to the same template set at both train and test time. This potential confound is not discussed.
- Improvements over RTB+RS are modest in most tasks, suggesting the contrastive loss provides meaningful benefit only in specific regimes. A clearer characterization of when and why the proposed method outperforms reward shaping would strengthen the paper's contribution.

---

> ### Author Rebuttal · Authors · 2026-03-31
>
> Thank you for the valuable comments!
>
> ## Advantages of GFlowNet objectives
> We agree that in the SMILES sequence generation setting, the trajectory-to-terminal mapping is injective and the TB/RTB objective reduces to PCL, as noted in Sec 2.1. Our claim is not that GFN provides a fundamentally different objective from other MaxEnt RL, nor that it is universally superior to simpler autoregressive alternatives.
>
> The main reason we adopt the GFN is its natural compatibility with our off-policy post-training algorithm. In particular, we propose replay training with separate pos/neg buffers, where pos samples are reused for RTB updates and neg samples are incorporated via the contrastive auxiliary loss. This off-policy training recipe is a central contribution, and it is more naturally expressed within the GFN framework than in standard PPO/REINFORCE-style MaxEnt RL (on-policy). Furthermore, our replay-based training algorithm is not limited to injective GFN MDPs, but could be applied to generalized cases too.
>
> ## Training synthesizability oracle
>
> In our view, **the use of separate synthesizability oracles for evaluation (AiZynthFinder/USPTO and SA score) does not artificially favor S3-GFN**. We report results both under the training-time evaluator (“Positive”) and under external evaluators. The latter are well-established reaction-based synthesizability metrics rather than being introduced specifically for our method.
>
> In fact, reaction-based baselines are naturally strongest on the training-time criterion itself, since their positive ratio is 1 by design. On the other hand, the external evaluators are used to test whether the model can generalize beyond the specific oracle used during training, rather than only under one hard-coded synthesis definition. Note that we directly adapt the external evaluator setup used in the previous reaction-based works. We will clarify this point more clearly in the revision.
>
> **Sensitivity test on synth evaluator**: we also provide results with the Enamine REAL reaction set as the training-time oracle, which show similar overall trends.
>
> | | Pos | Pos. Top100 | Avg. sEH | Div. | SA | AiZynth |
> | - | - | - | - | - | - | - |
> | S3-GFN (REAL) | 0.953 ± 0.009 | 1.054 ± 0.001 | 1.023 ± 0.002 | 0.757 ± 0.004 | 2.405 ± 0.013 | 0.980 ± 0.014|
>
> ## Comparison with RTB+RS
>
> The advantages of S3-GFN over RTB+RS are modest in Table 1 because this setting is relatively easy. However, **S3-GFN's benefits become more visible in the practical regimes that motivate our method**: under constraint changes and sample-efficient settings.
>
> Under constraint changes (Sec 5.2), S3-GFN preserves substantially higher uniqueness with better Pos. Top100.  Also, in the sample-limited setting (Sec 5.3), RTB+RS degrades significantly and suffers from early convergence.
>
> As you suggested, we conduct additional experiments under heavier constraint changes, which shows clearer advantanges.
>
> **With 15 reactions**: # of pos samples in the buffer is 2241
>
> | | Zeroshot | RTB+RS | S3GFN |
> | - | - | - | - |
> | Avg. SEH | 1.010 ± 0.001 | 0.943 ± 0.001 | 1.007 ± 0.001 |
> | Pos. Ratio | 0.620 ± 0.023 |  0.865 ± 0.020 | 0.838 ± 0.012 |
> | # Unique | 923.7 ± 3.1 | 795.0 ± 8.5 | 861.3 ± 13.7 |
> | Pos. Top100 | 1.035 ± 0.002 | 1.029 ± 0.001 | 1.037 ± 0.000 |
>
> **With 10 reactions**: # of pos samples in the buffer is 1073
>
> | | Zeroshot | RTB+RS | S3GFN |
> | - | - | - | - |
> | Avg. SEH | 1.010 ± 0.001 | 0.862 ± 0.042 | 1.007 ± 0.002 |
> | Pos. Ratio | 0.174 ± 0.018 | 0.637 ± 0.061 | 0.844 ± 0.010 |
> | # Unique | 923.7 ± 3.1 | 550.7 ± 22.4 | 687.3 ± 14.4 |
> | Pos. Top100 | 1.035 ± 0.002 | 1.016 ± 0.004 | 1.038 ± 0.000 |
>
> To further explain this behavior, we provide a new gradient analysis, which is also consistent with our empirical observations. In S3-GFN, for each positive sample, the gradient coefficients are
> $\frac{\partial \ell_i}{\partial s_\theta(\tau_i^+)}=-\frac{S_-}{e^{s_\theta(\tau_i^+)}+S_-}, \frac{\partial \ell_i}{\partial s_\theta(\tau_j^-)}=\frac{e^{s_\theta(\tau_j^-)}}{e^{s_\theta(\tau_i^+)}+S_-},$ where $S_- := \sum_{\tau_j^-\in B^-} e^{s_\theta(\tau_j^-)}$.
> Thus, when $e^{s_\theta(\tau_i^+)} \gg S_-$ (meaning that positive and negative samples are already sufficiently separated), both coefficients are close to zero, implying $\nabla_\theta L_{aux}\approx 0$.
>
> In contrast, under reward shaping, negative samples remain inside the main RTB objective itself, so their gradients do not disappear simply because they are already separated from positives. We will include this new analysis in the manuscript.
>
> ## Mutation-based negative samples
>
> We'd like to clarify that **mutation-based negatives are not the main source of negative samples** in our method; **negatives are mainly collected from the current training policy**. Mutation-based local negatives are suggested as a supplemental strategy when the policy generates very few negative samples, which can happen as training shifts probability mass toward the synthesizable region.

---

> > ### Author Rebuttal · Reviewer_npKo · 2026-04-02
> >
> > I thank the authors for the detailed rebuttal. The additional experiments under heavier constraint shifts and the gradient analysis were particularly helpful. Below I summarize the status of each concern.
> > ### Q1 (GFlowNet vs MaxEnt RL): Unresolved.
> > The authors acknowledge that "our claim is not that GFN provides a fundamentally different objective from other MaxEnt RL." In the injective SMILES setting, TB/RTB reduces to PCL, which is functionally equivalent to MaxEnt RL. The stated advantage of GFlowNet, namely off-policy compatibility, is not unique to the framework; off-policy replay is standard in MaxEnt RL (e.g., SAC). The argument that the method "could be applied to generalized (non-injective) cases" is valid in principle, but this paper exclusively operates in the injective setting. Moreover, the method's core structure, RTB with a pretrained SMILES prior, is itself fundamentally tied to the injective setting: learning a meaningful prior distribution requires autoregressive language modeling over large-scale SMILES corpora, which is only straightforward when the trajectory-to-molecule mapping is injective. In non-injective MDPs (e.g., fragment-based or reaction-based), the same molecule can be reached through multiple trajectories, making prior learning substantially more difficult, and large-scale trajectory corpora for such MDPs are generally unavailable. Therefore, the generalization argument does not hold in practice for this method. Without an empirical comparison against MaxEnt RL or autoregressive fine-tuning under the same contrastive constraint formulation, the necessity of the GFlowNet framework remains unjustified.
> > ### Q2 (Oracle mismatch): Partially resolved.
> > The additional experiment with the Enamine REAL reaction set as the training oracle is appreciated and shows similar trends. However, the structural asymmetry in the evaluation setup remains: reaction-based baselines are inherently constrained to their training template set and cannot generalize beyond it, while S3-GFN operates in the full SMILES space. The high AiZynthFinder success rates in Table 2 may therefore reflect the flexibility of the SMILES representation rather than a property of the proposed training method. This does not invalidate the results, but the comparison is not fully controlled.
> > ### Q3 (Improvements over RTB+RS): Largely resolved.
> > The heavier constraint shift experiments (15 and 10 reactions) convincingly demonstrate that S3-GFN's advantages become pronounced under more restrictive constraints. In particular, at 10 reactions, S3-GFN achieves substantially higher positive ratio (0.844 vs 0.637) and uniqueness (687 vs 551) while maintaining better Top-100 sEH. The gradient analysis explaining the natural saturation of the auxiliary loss, in contrast to the sustained suppression under reward shaping, provides a clear mechanistic explanation.
> > ### Q4 (Negative sample quality): Partially resolved.
> > The clarification that mutation-based negatives are supplemental and that negatives are primarily collected from the current training policy is helpful. However, as training progresses and the policy shifts toward the synthesizable region, the flow of new negatives diminishes. The behavior of the method when the negative buffer becomes stale or imbalanced (raised in Q4 of the original review) was not empirically characterized.
> >
> > Given the above, my core concern regarding the necessity of the GFlowNet framework (Q1) remains unresolved. While the empirical results are solid and the practical value of soft-constrained synthesizable generation is acknowledged, the methodological contribution is difficult to assess without a comparison against simpler baselines (e.g., MaxEnt RL) under the same constraint formulation. I maintain my current score.

---

> > > ### Author Response · Authors · 2026-04-06
> > >
> > > We believe that the reviewer and we agree on several key points, but there may still be a difference in framing about what the paper is intended to claim.
> > >
> > > Our main points are the following:
> > >
> > > * An injective SMILES MDP with a pretrained SMILES prior is a practically effective design choice, compared with the previously standard direction of using non-injective hard-constrained reaction-based MDPs without such a prior.
> > > * Our main contribution is the soft-constrained off-policy post-training framework: replay-based training with separate positive/negative usage and an auxiliary contrastive loss.
> > >
> > > We do **not** claim that GFlowNet is fundamentally different from, or generally better than, off-policy MaxEnt RL in the injective setting; in this case, they are essentially identical [1,2,3]. The reason we use the name S3-GFN is to emphasize the contrast with representative baselines such as SynFlowNet, which rely on non-injective hard-constrained MDPs. Our intended emphasis is therefore on the injective SMILES formulation and soft-constrained off-policy training, rather than on an objective-level claim that GFlowNets are uniquely superior in this setting.
> > >
> > >
> > > Also, we did not intend to claim practical extension to non-injective settings (e.g., graph-based generation). Our earlier statement was only meant in a framework-level sense: the auxiliary loss and the separate replay design are not inherently tied to the injective setting. The practical recipe studied in this paper is specifically focused on the injective SMILES setting.
> > >
> > > Following the reviewer’s suggestion, we additionally implemented Soft Q-Learning (SQL) and Soft Actor-Critic (SAC) under the same off-policy soft-constrained training scheme (same SMILES fine-tuning setup, same auxiliary loss, and same hyperparameter budget). We also implement Log-Partition Variance Gradient (VarGrad) [4, 5] objective which is trajectory-level contraint matching loss for GFlowNets smiliar with RTB.
> > >
> > >
> > > **AUC-Top10 with three independent runs**
> > > |  | Sum over 23 tasks |
> > > | - | - |
> > > | RTB + ours | 12.686 |
> > > | VarGrad + ours | 11.542 |
> > > | SQL + ours | 10.655 |
> > > | SAC + ours | 9.781 |
> > >
> > >
> > > S3-GFN performed better than these alternatives. Importantly, we **do not interpret this as evidence that GFlowNet (RTB, VarGrad) is generally better than MaxEnt RL (SQL, SAC)**. Rather, we believe trajectory-level matching (RTB, VarGrad) is more effective than local matching (SQL, SAC) in this sparse-reward combinatorial setting. Since rewards are obtained only after completing the full trajectory, local value estimation and credit assignment are much harder in this regime.
> > >
> > >
> > > Regarding the oracle-mismatch concern, we do not view the use of external evaluators as unfair or unsystematic. **We follow the same external evaluation protocol used in prior reaction-based work** [6,7], and we also report the training-time metric itself. From our perspective, this evaluation **highlights an important limitation of hard-constrained reaction-based GFlowNets**: they are tied to a specific synthesizability definition by construction. This guarantees synthesizability under that exact condition, but can be brittle under different evaluators. In this sense, the stronger transfer of S3-GFN under different evaluators is part of the intended comparison, rather than an uncontrolled confound. We view this as a practical advantage of our formulation, especially since synthesizability definitions can vary, and available evaluators may themselves be overly theoretical or vendor-specific.
> > >
> > >
> > > -------
> > >
> > > [1] Tiapkin et al. "Generative flow networks as entropy-regularized RL." AISTATS (2024).
> > >
> > > [2] Deleu et al. "Discrete probabilistic inference as control in multi-path environments." UAI (2024).
> > >
> > > [3] Deleu et al. "Relative trajectory balance is equivalent to Trust-PCL." arXiv preprint (2025).
> > >
> > > [4] Lorenz et al. "Vargrad: a low-variance gradient estimator for variational inference." NIPS (2020)
> > >
> > > [5] Zhang et al., "Robust Scheduling with GFlowNets" ICLR (2023)
> > >
> > > [6] Cretu et al. "Synflownet: Design of diverse and novel molecules with synthesis constraints." ICLR (2025).
> > >
> > > [7] Seo et al. "Generative flows on synthetic pathway for drug design." ICLR (2025).

---

### Official Review · Reviewer_w2xB · 2026-03-11

**Soundness:** 2
**Presentation:** 2
**Significance:** 2
**Originality:** 3
**Overall Recommendation:** 3
**Confidence:** 4

**Summary:**

The authors propose S3-GFN, a sequence-based Generative Flow Network (GFlowNet) designed to generate synthesizable SMILES strings via soft regularization. To bypass the scalability and flexibility issues of hard-constrained methods (which rely on predefined reaction templates and building blocks), S3-GFN leverages a pre-trained SMILES prior to generate candidate molecules. It then uses an external synthesizability oracle (such as AiZynthFinder) to label these candidates, creating buffers of positive (synthesizable) and negative (unsynthesizable) samples. The model is trained using a trajectory balance objective to fit the distribution of synthesizable molecules, alongside an auxiliary contrastive loss designed to push apart the distributions of synthesizable and unsynthesizable samples. Ultimately, S3-GFN aims to guide molecular generation toward high-reward, synthesizable chemical spaces.

**Compliance With Llm Reviewing Policy:**

Affirmed.

**Final Justification:**

I appreciate the authors' response, but my primary concern remains. S3-GFN treats synthesizability as a distribution-fitting task based on external labels, rather than generating an actual synthetic process.  Without explicit pathways, the results remain difficult for chemists to interpret or implement, limiting the model's practical utility.

**Key Questions For Authors:**

- Table 1 indicates that adding negative samples ($L_{aux}$) yields negligible gains over the RTB + RS baseline and significantly degrades the 'Positive' metric. Can you clarify the empirical value of this contrastive mechanism? Are there specific settings where its benefits are more pronounced, or is the model primarily driven by the prior and RTB?


- How do you justify evaluating the model's synthesizability using the exact same oracle (AiZynthFinder) used to label the training data?

- What is the rationale for reporting results for only 5 specific subsets of the PCBA dataset in Table 2? Could you provide the results for all 15 subsets, or the overall mean performance, to better demonstrate generalizability?

- Can you provide precise mathematical definitions, or at least standardized literature references, for all the evaluation metrics used in the experiments to ensure clarity and reproducibility?

- Regarding the claim in Section 5.1.1 that the model yields favorable properties like drug-likeness inherited from the prior: did you attempt to explicitly measure these properties across the generated distribution, or jointly optimize for them (e.g., QED), to substantiate this claim?

**Limitations:**

- The authors must explicitly acknowledge that their method relies on an oracle (AiZynthFinder) that only identifies theoretical synthesis pathways. Furthermore, they should clearly state as a limitation that their model does not account for practical laboratory constraints, such as reaction yield, financial cost, reagent availability, and reaction conditions.

- The authors should discuss that by moving away from hard-constrained methods, S3-GFN loses the ability to generate explicit, step-by-step synthesis routes . This makes the model's output less interpretable for bench chemists compared to methods like SynFlowNet.

**Strengths And Weaknesses:**

Strengths:

- The paper addresses a highly relevant and historically challenging bottleneck in AI-driven drug discovery: bridging the gap between computational molecular generation and practical laboratory synthesis.

- The shift from hard-constrained, template-based state/action spaces to a soft-regularized GFlowNet is an elegant approach. This allows for much greater flexibility and scalability when exploring vast chemical spaces.


Weaknesses:

- Unlike methods such as SynFlowNet , which generate molecules alongside explicit, step-by-step synthesis routes to guarantee synthesizability, S3-GFN does not produce synthesis pathways. It merely learns to fit the distribution of molecules labeled "synthesizable" by an external oracle. Consequently, its synthesizability stems entirely from imitating a black box, failing to explain what specific chemical patterns the model has actually learned and lacking explicit theoretical guarantees.

- A primary methodological contribution is the contrastive learning loss ($L_{aux}$) designed to separate synthesizable and unsynthesizable distributions. However, Table 1 shows that incorporating negative samples yields negligible improvements. Compared to the RTB + RS baseline (which uses no negative samples), S3-GFN only improves Pos. Top100 sEH and sEH by 0.003 and 0.006, respectively, while the 'Positive' metric suffers a significant drop of 0.042. This suggests the proposed contrastive mechanism is empirically weak.

- The model relies heavily on AiZynthFinder to define ground-truth synthesizability during training, yet evaluates performance using the "AiZynthFinder Success Rate" (Table 2). This evaluation is circular. Furthermore, AiZynthFinder only identifies theoretical synthesis pathways. It completely ignores practical laboratory constraints such as reaction difficulty, chemical yield, and financial cost. This leaves a massive gap between "theoretical" and "practical" synthesizability that the paper fails to address.

- In Table 2, results are reported for only 5 subsets of the PCBA dataset without providing a rationale for this selection. To properly demonstrate the model's robustness and generalizability, the authors should report results for all 15 subsets or, at the very least, provide the mean performance across the entire dataset.

- In Section 5.1.1, the authors claim the model yields "favorable chemical properties likely inherited from the prior, such as high drug-likeness." However, S3-GFN does not explicitly optimize for these properties. It would strengthen the paper to see if the authors attempted to jointly optimize synthesizability alongside other specific properties (e.g., QED or drug-likeness), beyond the joint activity optimization mentioned in Section 5.1.2.

- The experiments utilize a large number of metrics, but there is a distinct lack of concrete definitions or mathematical explanations for these metrics in both the main text and the appendix. This creates unnecessary confusion and severely hinders reproducibility.

---

> ### Author Rebuttal · Authors · 2026-03-31
>
> Thanks for the valuable comments!
>
> ## Synthesis pathway generaion and intepertation
>
>
> We agree that S3-GFN does not perform step-by-step route generation. However, this does not mean that it is disconnected from synthesis routes. Rather than directly parameterizing a route distribution $P_F(\rho)$, S3-GFN learns a terminal molecule distribution $p_\theta(x)$ over positive molecules, where positivity is defined by the existence of a valid synthetic pathway under the given reaction-template and building-block system.
>
> A route distribution can still be induced afterward as $P_F(\rho) \approx p_\theta(x) Q_B(\rho \mid x)$, where $Q_B(\rho \mid x)$ is a retrosynthesis or rule-based backtracking procedure conditioned on $x$. In this sense, chemical patterns **remain interpretable** at the terminal-molecule level, while route-level structure can be recovered in a decoupled second stage, as shown in Fig. 4. In addition, the chemistry literature already provides many effective retrosynthesis tools, which can be naturally combined with the inductive bias encouraged by S3-GFN.
>
> We also agree that soft-constraint training does not provide the same constructive theoretical guarantee as route-explicit generation. However, we do not see this as a fundamental weakness, but as a familiar trade-off in constrained optimization: many successful methods relax hard feasibility during construction to obtain smoother and more effective optimization. Our method follows the same principle, trading strict route-level guarantees for greater flexibility, transferability, and practical optimization performance.
>
>
>
>
> ## Regarding synthesizability evaluators
>
> **AiZynthFinder is not used during training**; it is only used as an external evaluator, together with SA score. During training, we define X' using the same template/building-block criterion as prior reaction-based work for direct comparison.
>
> We agree that existing synthesizability criteria are often imperfect or overly theoretical, so relying on a single criterion can be risky. This is exactly why we use a soft-constrained setup that decouples molecule generation from synthesizability evaluation.
>
> This gives two advantages:
> - S3-GFN performs well not only under the training-time criterion (“Positive”), but also under external evaluators, suggesting that it is **not tied to one specific synthesizability definition**;
> - The framework is **modular and flexible**. If more practical synthesizability evaluators are became available, they can be incorporated without changing the generation process.
>
> We also provide new results using the Enamine REAL reaction set as the training-time evaluator, which show similar overall trends (see the provided link below).
>
> At the same time, we acknowledge that current evaluators do not fully capture practical laboratory constraints. We will state this limitation more explicitly and discuss it in the conclusion.
>
>
>
> ## Comparison with RTB + RS
> While a high positive ratio is important, it **should not be achieved by becoming overly conservative and collapsing to a small safe region**, since this can exclude promising high-reward candidates. For this reason, we also consider **Pos. Top100 sEH**. While the gain is modest in some setting, S3-GFN consistly achieves better Pos. Top100 sEH, meaning that it **preserves stronger synthesizable candidates instead of collapsing toward safer but less rewarding regions**. The advantage becomes **much clearer in the practical regimes** that motivate our method. Under constraint changes (Sec 5.2), S3-GFN preserves higher uniqueness with better Pos. Top100 sEH. In the sample-limited setting (Sec 5.3), RTB+RS degrades significangly, whereas S3-GFN shows strong sample efficiency.
>
>
> ## Role of the prior
> The pretrained prior already captures rich chemical knowledge from large molecular data, which induces broadly reasonable properties such as high QED. RTB helps preserve these prior-induced properties by discouraging unnecessary deviation from the prior, even without explicitly optimizing them; see the additional results. However, the prior alone is not sufficient, since it does not yield a sufficiently high fraction of molecules satisfying the target synthesizability criterion. Our method therefore improves synthesizability control while preserving the useful chemistry encoded by the prior.
>
>
> ## Additional results
> We provide additional results at https://anonymous.4open.science/r/S3GFN/additional.pdf
> - We use 5 subsets mainly reported in RxnFlow, but we provide additional results on PCBA
> - Joint optimization of sEH + QED: this further improves QED while introducing the standard multi-objective trade-off with sEH, whereas optimizing only sEH largely preserves the prior’s QED level.
>
>
> ## Reproducibility
> We provided the code for reproducibility. As suggested, we will add a precise definition of the metric in the revision. Most evaluation metrics are directly adopted from the corresponding benchmark, as noted in each section.

---

### Official Review · Reviewer_FPNa · 2026-03-11

**Soundness:** 3
**Presentation:** 4
**Significance:** 3
**Originality:** 3
**Overall Recommendation:** 5
**Confidence:** 4

**Summary:**

The authors propose S3-GFN, which builds on a pretrained SMILES language model prior and performs GFlowNet post-training with Relative Trajectory Balance (RTB) and a contrastive auxiliary loss that separates synthesizable and unsynthesizable trajectories using replay buffers. The authors validate the method on several molecular design tasks, including proxy optimization for sEH binding affinity and SBDD tasks using docking. The method shows high synthesizability while maintaining or improving rewards compared to a reaction-based GFN baseline, and also compares to standard reward shaping.

**Compliance With Llm Reviewing Policy:**

Affirmed.

**Final Justification:**

The authors addressed my key concerns during rebuttal, especially by adding the comparison to Saturn which I believe is a very relevant baseline for the proposed model, and clarifying how the set of synthesizable molecules ($X'$) is defined (I think this is very relevant information, and the procedure used is non-trivial, therefore I recommend the authors to include it in the next version of the manuscript). They also add valuable tests showing improvements over reward shaping under harder constraints. I will raise the significance score from 2 to 3 and the overall score from 3 to 5.

**Key Questions For Authors:**

- How are positive molecules being selected? Is this done via retrosynthesis? If yes, what are the reactions and building blocks being used?
- Could the authors comment on sample efficiency and how it compares to other methods?

**Limitations:**

Yes

**Strengths And Weaknesses:**

Strengths

- The central contribution of the work is replacing hard-constraint reaction-based MDPs with soft constraint regularization. S3-GFN keeps a simple SMILES MDP and learns synthesizability through distributional shaping. This is a really nice finding for the field of synthesizable molecule design, as it is *one of the first* works that shows that one can generate synthesizable molecules without relying on reaction-based molecule representations, but by relying on steering generation towards distributions of synthesizable molecules.
- The work nicely leverages RTB-based post-training, treating a pre-trained SMILES model as prior and learning the posterior proportional to $R(x)p_{prior}(x)$. The contrastive auxiliary loss during replay training is technically simple and fits well with the off-policy training paradigm, showing improved control over synthesizable molecule generation.
- The paper shows adaptation capabilities to revised chemical reactions

Questions / Weaknesses

- It is not clear whether $X’$ is defined from the 105 reactions in SynFlowNet and the Enamine stock or using AiZynthFinder. If it is the former, the paper lacks details around how retrosynthesis is set up in such a way to use these custom reactions and building blocks. If the latter is true (or if the retrosynthesis model also uses AiZynthfinder default reactions and blocks, on top of SFN reactions), then the results in Figure 3 and Table 2 on AiZynthfinder are inflated, as the model directly optimizes for AiZynthfinder synthesizability, which is unfair for the baselines.
- The model compares, among other baselines, against SynFlowNet which is a suitable baseline to assess MDP design and training/post-training recipe, but does not compare against [1] by Jeff Guo et al. which I believe is the next most relevant baseline as it also uses a SMILES-based model and learning signal from a retrosynthesis oracle that assesses compounds as synthesizable and non-synthesizable (albeit with a different algorithm). I believe this comparison is crucial.
- The performance under constraint changes in Figure 3 shows that there is little to no benefit over reward shaping.
- A mode exploration analysis is lacking. Does S3-GFN preserve high mode exploration compared to baselines and among different experiments in the paper?

If the points above are addressed, I will consider raising my score.


[1] Jeff Guo and Philippe Schwaller, Directly Optimizing for Synthesizability in Generative Molecular Design using Retrosynthesis Models, 2024, arXiv, 2407.12186

---

> ### Author Rebuttal · Authors · 2026-03-31
>
> Thanks for the insightful reviews. We are addressing your comments one by one below.
>
> ## **Training synthesizability constraints**
>
> Thank you for pointing this out. For direct comparison with SynFlowNet, we define X' as the set of molecules that admit a valid synthetic pathway under **the same 105 reaction templates and Enamine stock library used in SynFlowNet**. **AiZynthFinder is not used to define X'** and is not used anywhere in training; it is used only as an external evaluation metric. Therefore, there is no training-time leakage or direct optimization toward AiZynthFinder synthesizability. We agree that the retrosynthesis setup used to determine membership in X' was not described clearly enough in the current manuscript, and we will add these implementation details explicitly in the revised version.
>
> ## **Baseline by Guo et al. [1]**
> We agree that your suggested baseline, Saturn [1] is highly relevant. Similar to our method, Saturn uses a SMILES-based model and a retrosynthesis-based synthesizability signal. It uses REINVENT with reward shaping, therefore it share the same methodology with the REINVENT + RS baseline considered in Section 5.3.
>
> Following your suggestion, we additionally include Saturn results below on the sEH task and the sample-efficient benchmark, which are the main benchmarks in our paper. We use the original Saturn codebase with only minimal modifications required to adapt it to these two tasks.
>
> **sEH**
> |  | Positive | Avg. sEH | Pos. Top100 sEH|
> | - | - | - | - |
> | Saturn | 0.885 ± 0.020 | 0.906 ± 0.018 | 1.027 ± 0.002 |
> | S3GFN | 0.945 ± 0.009 | 1.009 ± 0.000 | 1.043 ± 0.001 |
>
>
> **Sample efficiency**
> |  | Total AUC-Top10 |
> | - | - |
> | Saturn | 13.548 |
> | S3GFN | 12.686 |
> | S3GFN (Genetic Expl.) | 14.255 |
>
>
> S3-GFN achieves a higher positive ratio, average sEH, and Positive Top-100 sEH on the sEH task. In addition, S3-GFN with genetic off-policy exploration attains better sample efficiency on the PMO benchmark (14.255 vs. 13.548 for Saturn). While both methods benefit from strong SMILES-based representations, we attribute the gap mainly to S3-GFN’s off-policy design: positive/negative replay improves feasibility alignment, and genetic off-policy exploration improves sample-efficient reward optimization. In contrast, Saturn uses an on-policy objective and is therefore less suited to leveraging these off-policy mechanisms.
>
> We will include these results, along with the detailed experimental setup and implementation details, in the revised manuscript.
>
>
> ## **Performance under constraint shifts**
>
> We agree that the performance difference between S3-GFN and RTB + RS under constraint changes seems modest. The current task is relatively easy: even after the constraint change, it is still possible to find many high-reward molecules, so the margin between methods is naturally small. However, even in this case, **S3-GFN preserves higher diversity (significantly more unique molecules) while achieving better Pos. Top100**.
>
> To make this comparison more clear, we additionally considered heavier constraint changes by reducing the allowed reaction subset to 15 and 10 reactions from 32. In these settings, the positive ratio before realignment is much lower.
>
> **With 15 reactions**
>
> | | Zeroshot | RTB+RS | S3GFN |
> | - | - | - | - |
> | Avg. SEH | 1.010 ± 0.001 | 0.943 ± 0.001 | 1.007 ± 0.001 |
> | Pos. Ratio | 0.620 ± 0.023 |  0.865 ± 0.020 | 0.838 ± 0.012 |
> | # Unique | 923.7 ± 3.1 | 795.0 ± 8.5 | 861.3 ± 13.7 |
> | Pos. Top100 | 1.035 ± 0.002 | 1.029 ± 0.001 | 1.037 ± 0.000 |
>
>
> **With 10 reactions**
>
> | | Zeroshot | RTB+RS | S3GFN |
> | - | - | - | - |
> | Avg. SEH | 1.010 ± 0.001 | 0.862 ± 0.042 | 1.007 ± 0.002 |
> | Pos. Ratio | 0.174 ± 0.018 | 0.637 ± 0.061 | 0.844 ± 0.010 |
> | # Unique | 923.7 ± 3.1 | 550.7 ± 22.4 | 687.3 ± 14.4 |
> | Pos. Top100 | 1.035 ± 0.002 | 1.016 ± 0.004 | 1.038 ± 0.000 |
>
> In simpler settings, **RTB+RS can appear competitive, but under stronger constraint shifts it becomes more conservative and loses both diversity and high-reward candidates**. In contrast, S3-GFN better preserves exploration while retaining strong candidates within the new synthesizable space.
>
> ## **Mode exploration**
> **We have analyzed mode exploration by computing all metrics on the Top-100 _diverse_ candidates** in Sec 5.1.2. The top 100 high-reward candidates that are mutually distinct under a predefined dissimilarity threshold, that is **100 distinct modes.**
>
> In particular, Table 2 reports vina score and synthesizability of **Top-100 diverse candidates** on pocket-specific optimization tasks following the setup from RxnFlow; see Appendix B.3. In our case, this is computed over the positive samples, i.e., molecules that satisfy our training-time synthesizability criterion. Therefore, the results in Table 2 show that **S3-GFN can preserve high mode exploration within the synthesizable space**. We will revise the manuscript to clarify this point.

---

> > ### Author Rebuttal · Reviewer_FPNa · 2026-04-02
> >
> > Thank you for the answers. Please see remaining concerns below:
> >
> > **Constraints**: Thank you for clarifying that the 105 templates and Enamine stock from SFN were used. Still, what is the procedure under which X' was defined? How was decomposition performed? Is it via a retrosynthesis software or how do you ensure that the molecules are synthesizable under this vocabulary?
> >
> > **Mode exploration**: Thanks for clarifying. I understand that metrics are reported over 100 distinct molecules, but this is not a direct report of the diversity of the samples generated by the model, and does not *explicitly* reflect the mode exploration capabilities of the model. Instead, I am suggesting sampling 1k /10k molecules and reporting mode exploration and diversity metrics.

---

> > > ### Author Response · Authors · 2026-04-03
> > >
> > > Thank you for sharing your remaining concerns. We are glad that several of the earlier points were clarified.
> > >
> > > ## On how X' is defined / retrosynthesis procedure
> > > X' is defined as the **set of molecules for which a valid synthetic pathway can be found via a retrosynthesis search**. We employ the retrosynthesis search from RxnFlow (one of the reaction-based GFlowNets), where it is used as a rule-based backward policy in GFlowNet. Starting from the target molecule, the search recursively applies reverse reactions and checks whether the molecule can be decomposed into purchasable building blocks within the allowed number of steps (at most 3 synthesis steps).
> > >
> > > ## On mode exploration
> > > We report **diversity over 1,000 generated molecules in Table 1**, where S3-GFN is slightly lower than SynFlowNet (0.764 vs. 0.801). However, Table 6 in Appendix C.2  shows that this reward-diversity trade-off is controllable, which is a well-known property of GFlowNets. S3-GFN achieves a higher sEH score while maintaining similar diversity (0.803). So the results show that **S3-GFN can preserve strong mode exploration while targeting high-reward synthesizable molecules**.
> > >
> > > For Table 2, we additionally report diversity for 1k generated samples, although direct comparison is limited since the original work does not provide the corresponding diversity metric.
> > >
> > > |  | ADRB2 | ALDH1 | ESR ago | ESR antago | FEN1 |
> > > | - | - | - | - | - | - |
> > > | Div. | 0.783 ± 0.001 | 0.796 ± 0.002 | 0.786 ± 0.002 | 0.796 ± 0.003 | 0.794 ± 0.002 |

---

### Decision · Program_Chairs · 2026-04-30

**Decision:**

Accept (regular)

**Comment:**

The paper proposes an improvement for generative GFlowNet-based models for molecules which applies post-training with soft synthesizatibility constraints instead of the coarse hard ones that have been applied to date. This is achieved using contrastive learning.

Two reviewers recommend weak accept / accept while there is one weak reject. All acknowledge that the empirical results are solid and recognize the practical and technical value of soft-constrained synthesizable generation. Some of the reviewers remaining concerns seem out of scope (e.g., MaxEnt RL discussion). The "concern" that the approach does not provide synthesis routes is an obvious consequence of the model design. Furthermore, the high AiZynthFinder Success Rate shows that route proposals can easily be obtained in this way. I want to note that the reaction templates used in this paper (from SynFlowNet) overlap with the ones used in AiZynthFinder. So there is connection between training and evaluation. However, these are the ones commonly used in retrosynthesis prediction to date.